# CLASP2 binding to curved microtubule tips promotes flux and stabilizes kinetochore attachments

Hugo Girão[1,2], Naoyuki Okada[1,2], Tony A. Rodrigues[1,2,3], Alexandra O. Silva[1,2], Ana C. Figueiredo[1,2], Zaira Garcia[1,2], Tatiana Moutinho-Santos[1,2], Ikuko Hayashi[4], Jorge E. Azevedo[1,2,3], Sandra Macedo-Ribeiro[1,2], and Helder Maiato[1,2,5]

**CLASPs are conserved microtubule plus-end–tracking proteins that suppress microtubule catastrophes and independently localize to kinetochores during mitosis. Thus, CLASPs are ideally positioned to regulate kinetochore–microtubule dynamics required for chromosome segregation fidelity, but the underlying mechanism remains unknown. Here, we found that human CLASP2 exists predominantly as a monomer in solution, but it can self-associate through its C-terminal kinetochore-binding domain. Kinetochore localization was independent of self-association, and driving monomeric CLASP2 to kinetochores fully rescued normal kinetochore–microtubule dynamics, while partially sustaining mitosis. CLASP2 kinetochore localization, recognition of growing microtubule plus-ends through EB–protein interaction, and the ability to associate with curved microtubule protofilaments through TOG2 and TOG3 domains independently sustained normal spindle length, timely spindle assembly checkpoint satisfaction, chromosome congression, and faithful segregation. Measurements of kinetochore–microtubule half-life and poleward flux revealed that CLASP2 regulates kinetochore–microtubule dynamics by integrating distinctive microtubule-binding properties at the kinetochore–microtubule interface. We propose that kinetochore CLASP2 suppresses microtubule depolymerization and detachment by binding to curved protofilaments at microtubule plus-ends.**

## Introduction

The fine regulation of kinetochore (KT)–microtubule (MT) dynamics during mitosis ensures proper chromosome segregation by promoting error correction and spindle assembly checkpoint (SAC) satisfaction. MT dynamics are modulated throughout the cell cycle by several MT-associated proteins (MAPs; Maiato et al., 2004). Some MAPs specifically accumulate at the growing plus-ends of MTs and are collectively known as MT plus-end–tracking proteins or +TIPs (Akhmanova and Steinmetz, 2008). CLIP-associated proteins (CLASPs) are widely conserved +TIPs that stabilize MT plus-ends by suppressing catastrophes and promoting rescue (Aher et al., 2018; Al-Bassam et al., 2010; Lawrence et al., 2018; Majumdar et al., 2018). Humans have two CLASP paralogues, CLASP1 and CLASP2, which exist as different isoforms: α, β, and γ (Akhmanova et al., 2001; Inoue et al., 2000; Lemos et al., 2000).

CLASPs harbor three distinct functional domains: (1) a basic serine-rich region, also found in other +TIPs, comprising two serine–x-isoleucine–proline (SxIP) motifs that enable MT plus-end tracking via interaction with end-binding (EB) proteins (Honnappa et al., 2009; Mimori-Kiyosue et al., 2005); (2) two to three tumor overexpression gene (TOG) domains, which are structurally unable to bind to αβ-tubulin heterodimers present along straight protofilaments on polymerized MTs, and were proposed to recognize the curved conformation of MT plus-ends (Leano et al., 2013; Leano and Slep, 2019; Maki et al., 2015); and (3) a C-terminal domain (C-term; also known as CLIP-interacting domain) required for KT localization (Maia et al., 2012; Maiato et al., 2003a; Mimori-Kiyosue et al., 2006) and protein dimerization (Al-Bassam et al., 2010; Funk et al., 2014; Patel et al., 2012), as well as interaction with other KT proteins, including CLIP170, CENP-E, and Plk1 (Akhmanova et al., 2001; Dujardin et al., 1998; Maffini et al., 2009; Maia et al., 2012).

Mammalian CLASPs play redundant roles in the organization of the mitotic spindle, and interference with their function results in monopolar, short, and multipolar spindles (Logarinho et al., 2012; Maiato et al., 2003a; Maiato et al., 2003b; Mimori-Kiyosue et al., 2006; Pereira et al., 2006). CLASPs localize at the fibrous corona region of the KT throughout mitosis, where they play a critical role in the regulation of KT–MT dynamics required for MT poleward flux and turnover, as well as the correct alignment and segregation of chromosomes (Logarinho et al., 2012; Maffini et al., 2009; Maiato et al., 2003a; Maiato et al.,

...................................................................................................................................................................

[1]Instituto de Biologia Molecular e Celular, Universidade do Porto, Porto, Portugal; [2]Instituto de Investigação e Inovação em Saúde (i3S), Universidade do Porto, Porto, Portugal; [3]Instituto de Ciências Biomédicas Abel Salazar da Universidade do Porto, Porto, Portugal; [4]International Graduate School of Arts and Sciences, Yokohama City University, Yokohama, Japan; [5]Cell Division Group, Experimental Biology Unit, Department of Biomedicine, Faculdade de Medicina, Universidade do Porto, Porto, Portugal.

Correspondence to Helder Maiato: maiato@i3s.up.pt.

2005; Maiato et al., 2003b; Pereira et al., 2006). During prometaphase, CLASP1 interacts with the kinesin-13 Kif2B to promote KT–MT turnover, which is necessary for the correction of erroneous attachments (Maffini et al., 2009; Manning et al., 2010). As chromosomes bi-orient and cells transit into metaphase, CLASP1 interacts with Astrin, which promotes KT–MT stabilization required for SAC satisfaction (Manning et al., 2010). This places CLASP1 as part of a regulatory switch that enables the transition between labile-to-stable KT–MT attachments by establishing temporally distinct interactions with different partners at the KT. Additionally, CLASP2 phosphorylation by Cdk1 and Plk1 as cells progressively reach metaphase gradually stabilizes KT–MT attachments (Maia et al., 2012). While this broad picture provides important information about the molecular context in which CLASPs operate at the KT–MT interface, we still lack a detailed mechanistic view on how the intrinsic properties of CLASPs modulate KT–MT dynamics.

Here, we focused on human CLASP2 to investigate how its distinct functional domains affect mitosis, with emphasis on the regulation of KT–MT dynamics. Our findings revealed that KT CLASP2 integrates multiple independent features, including recognition of growing MT plus-ends through EB–protein interaction and the ability to associate with curved MT protofilaments through TOG2 and TOG3 domains to modulate KT–MT dynamics required for faithful chromosome segregation during mitosis in human cells.

## Results

### CLASP2 is a monomer in solution, but it can self-associate through its C-term

Previous studies have come to contradictory conclusions regarding the native oligomerization state of vertebrate CLASPs, with some works suggesting the formation of monomers (Emanuele et al., 2005; Drabek et al., 2006; Aher et al., 2018) or dimers in solution (Patel et al., 2012). CLASP orthologues in *Schizosaccharomyces pombe* and *Saccharomyces cerevisiae* also form homodimers in solution through their C-term, which was proposed to account for its KT function and binding to soluble αβ-tubulin heterodimers (Al-Bassam et al., 2010; Funk et al., 2014). However, whether this model applies to human CLASPs remains unknown.

To investigate this, we expressed and purified full-length human CLASP2α (CLASP2 FL) and determined its molecular mass using the Siegel and Monty method (Erickson, 2009; Siegel and Monty, 1966). This method combines the Stokes radius ($R_s$) obtained from size exclusion chromatography (SEC) and the sedimentation coefficient derived from density gradient centrifugation to determine the molecular mass of a protein. CLASP2 FL aggregated in low ionic strength buffers and interacted with the matrix of the SEC columns equilibrated with buffers containing 150 mM NaCl. Therefore, for $R_s$ determination of both recombinant and endogenous (see below) full-length CLASP2, buffers containing 400 mM NaCl were used. Accordingly, recombinant CLASP2 FL displayed an $R_s$ of 7.6 nm, a value comprised between that of ferritin ($R_s$ = 6.1 nm; molecular weight [MW] = 440 kD) and thyroglobulin ($R_s$ = 8.5 nm;

MW = 680 kD; Fig. 1, A and G). Sedimentation analyses revealed that CLASP2 FL displayed a sedimentation coefficient of 5.6 S (Fig. 1, B and G) and, applying the Siegel and Monty equation (Erickson, 2009), the calculated molecular mass of recombinant human CLASP2α is 179 kD, a value consistent with a monomeric state in solution. The large value of $R_s$ indicates that CLASP2 FL does not behave as a globular protein in solution, suggesting that it has a predominantly elongated shape and/or is rich in intrinsically disordered regions.

To assess potential differences in the oligomerization state between recombinant and endogenous CLASP2, an identical analysis was performed for endogenous CLASP2 in HeLa cells (Fig. 1, C and D). The hydrodynamic parameters obtained ($R_s$ = 7.6 nm; $s$ = 5.3 S) indicated that endogenous CLASP2 has an MW of 169.4 kD, a value equivalent to the one obtained for recombinant CLASP2α. Notably, although tubulin was present across the sucrose gradient and SEC, the determined molecular mass reveals that endogenous CLASP2 could not be recovered as part of a tubulin-containing complex. Only a very small fraction of CLASP2 and tubulin (most likely as polymerized MTs) was excluded from the SEC column (Fig. 1 C, lane 9). These data indicate that human CLASP2 exists predominantly as a monomer in solution and, unlike its yeast counterparts, it does not associate with soluble αβ-tubulin heterodimers, in agreement with previous reports (Aher et al., 2018; Maki et al., 2015).

To substantiate these results, we extended the determination of the molecular mass for a recombinant protein comprising amino acid residues Asp1213–Ser1515 of CLASP2α (CLASP2 C-term; theoretical MW of 38 kD), the domain putatively involved in homodimerization according to previous reports for human CLASP1 and its yeast counterparts (Al-Bassam et al., 2010; Funk et al., 2014; Patel et al., 2012). Combining SEC and sedimentation analysis, the molecular mass determined for CLASP2 C-term was 39 kD (Fig. 1, A, E, and G), indicating that under these experimental conditions (400 mM NaCl) CLASP2 C-term behaves as a monomeric protein. Contrary to CLASP2 FL (endogenous and recombinant), CLASP2 C-term did not interact with the SEC matrix at lower ionic strengths (Fig. 1 F), and the molecular mass obtained using 150 mM NaCl (34 kD; Fig. 1, A, F, and G) was also compatible with that of a monomer, regardless of the amount of protein loaded in the SEC column or the presence of the N- or C-terminal tags in CLASP2 C-term (Fig. S1).

Analysis of CLASP2 C-term by dynamic light scattering (DLS), which measures intensity fluctuations of scattered light resulting from Brownian motion of macromolecules in solution, allowed the determination of the diffusion coefficient (Dm) that correlates with the size of the particles in solution (Stetefeld et al., 2016). We found that CLASP2 C-term was monodisperse in solution, consisting of a single population (Fig. 1 H). Variations in Dm measured by DLS with increasing protein concentration can be used to estimate an interaction parameter that is negative or positive for attractive or repulsive protein interactions, respectively (Hassan et al., 2015; Li et al., 2004; Yadav et al., 2011). The plot of Dm with increasing concentration of CLASP2 C-term suggests the existence of intermolecular attractive interactions (Fig. 1 I). Considering that a weak

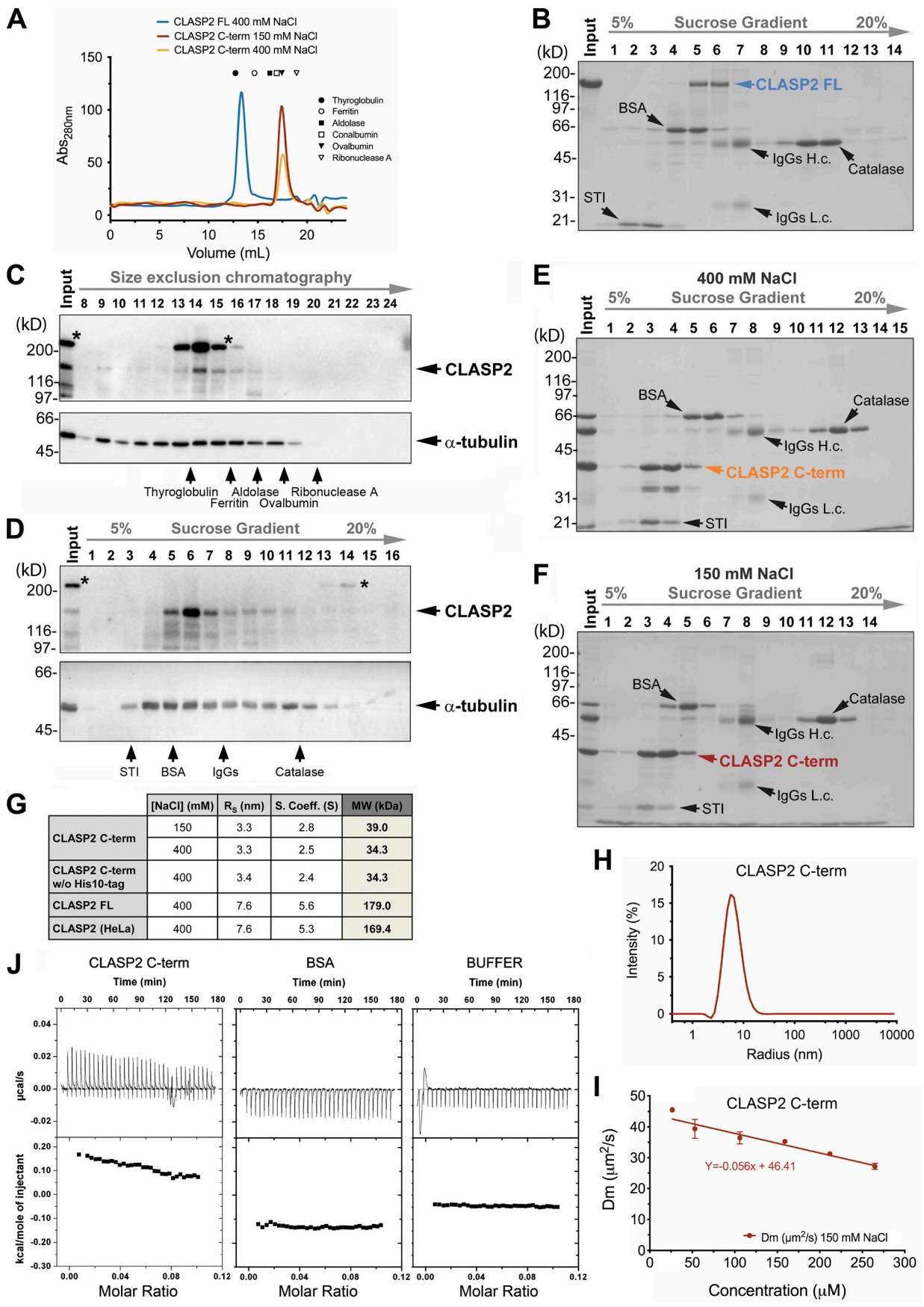

Figure 1. **CLASP2 is a monomer in solution, but it can self-associate through its C-term. (A)** Size exclusion chromatograms of recombinant human CLASP2 FL and CLASP2 C-term. The elution positions of the protein standards used to calibrate the column are shown. **(B)** Coomassie blue–stained SDS-PAGE showing the sedimentation behavior of recombinant human CLASP2 FL upon sucrose gradient centrifugation. The positions of catalase, IgGs, BSA, and STI added to the CLASP2 FL sample before centrifugation are indicated. Input, 4 μg of the recombinant CLASP2 FL protein analyzed in the gradient; IgGs H.c. and IgGs L.c., heavy and light chains of IgG, respectively. **(C and D)** Extracts from mitotic-enriched HeLa cells were subjected to SEC (C) and sucrose gradient centrifugation (D). Collected fractions were precipitated with TCA and analyzed by SDS-PAGE/Western blot with anti-CLASP2 and anti-tubulin antibodies (upper and lower panels, respectively). Input, 100 μg of mitotic-enriched HeLa cell extract. * indicates an unknown protein occasionally recognized by the anti-CLASP2 antibody. **(C)** Fraction 9 represents a void volume of the column. **(E and F)** Recombinant CLASP2 C-term was subjected to density gradient centrifugation in the presence of either 400 mM (E) or 150 mM (F) NaCl, as described above for CLASP2 FL. Input, mixture of recombinant CLASP2 C-term plus protein standards that were loaded on the sucrose gradients; IgGs H.c. and IgGs L.c., as in B. Numbers to the left in B–F indicate the molecular masses of the protein standards in kD. **(G)** Summary of $R_s$ (nm), S, and estimated MWs (kD) of the different CLASP2 proteins. **(H)** Analysis of recombinant CLASP2 C-term (26.5 μM) monodispersity by DLS. **(I)** Determination of recombinant CLASP2 C-term Dm by DLS. The linear regression extrapolated from the graphical representation of CLASP2 C-term Dm (μm²/s) at different concentrations (μM) allows the determination of the interaction parameter (−0.0012 μM⁻¹), demonstrating the existence of attractive intermolecular interactions. **(J)** Dilution ITC of CLASP2 C-term. The raw calorimetric dilution titration profile (top) and the integrated heat of dissociation (bottom) are shown for CLASP2 C-term, BSA, and ITC buffer, as indicated.

CLASP2 self-association ($K_D$ in the mid/high micromolar range) might prove elusive for the methods employed above, we analyzed a putative monomer–dimer equilibrium of CLASP2 C-term by dilution isothermal titration calorimetry (ITC; Pierce et al., 1999). Interestingly, dilution of CLASP2 C-term in buffer gave rise to endothermic heat pulses, consistent with molecular dissociation (Fig. 1 J). These data suggest a weak interaction, but the $K_D$ could not be reliably determined even at the maximum CLASP2 C-term concentration achievable (0.6 mM). Taking the volume of HeLa cells as a reference (1,100–2,600 μm³; Luciani et al., 2001) and the amount of CLASP2 they contain (4.0 fg/cell; Finka and Goloubinoff, 2013), the concentration of CLASP2 should be 9.4–22.2 nM, indicating that endogenous CLASP2 exists predominantly, if not only, as monomer. Nevertheless, CLASP2 self-association in cells might be context specific and depend on CLASP2 accumulation on particular cellular structures and interaction with other proteins (e.g., at KTs).

## CLASP2 C-term, but not self-association, is required and sufficient for KT localization

To investigate how CLASP2 impacts KT–MT dynamics during mitosis, we used an RNAi-based rescue system with RNAi-resistant, mRFP-CLASP2γ–derived constructs harboring specific mutations that disrupt the different CLASP2 functional domains (Maki et al., 2015), either alone or in combination (Fig. 2 A). Because CLASP2γ lacks the TOG1 domain normally present in the most abundant CLASP2α isoform, these constructs allowed us to directly test the importance for mitosis of the TOG1 domain, which bears structural similarity to the free tubulin-binding TOG domains of the XMAP215 family of MT polymerases (Ayaz et al., 2012; Leano and Slep, 2019; Nithianantham et al., 2018) and has been previously implicated in CLASPs binding to free αβ-tubulin heterodimers (Yu et al., 2016). In addition to the WT mRFP-CLASP2γ construct, we used a construct mutated in both TOG2 and TOG3 domains (2ea-3eeaa) that have an arched conformation that can only bind to αβ-tubulin heterodimers on curved MT protofilaments (Leano et al., 2013; Leano and Slep, 2019; Maki et al., 2015), a construct mutated in both SxIP motifs (IP12) that disrupts CLASP2-EB protein interaction and consequently MT plus-end tracking (Honnappa et al., 2009), and two constructs combining mutations at SxIP motifs and either

TOG2 or TOG3 domains (2ea-IP12 or IP12-3eeaa, respectively). Additionally, we used two other constructs: one with a truncated C-term (ΔC), and another with just the C-term, which mediates putative CLASP2 self-association (this study) and is required for KT localization (Maia et al., 2012; Mimori-Kiyosue et al., 2006). Finally, to dissect the requirement of putative CLASP2 self-association for its KT localization and function, we generated two additional constructs: one in which the human CLASP2 C-term was replaced by an artificial dimerization domain based on the Gcn4-leucine zipper sequence (ΔC-Gcn4), as shown previously for the *S. cerevisiae* CLASP orthologue Stu1 (Funk et al., 2014), and another construct in which the human CLASP2 C-term was replaced by Spc25 (ΔC-Spc25), a small monomeric KT protein that is part of the Ndc80 complex required for end-on KT–MT attachments (McCleland et al., 2004; Wigge and Kilmartin, 2001).

After stable transduction using a lentiviral system in a human U2OS cell line stably expressing photoactivatable (PA) GFP–α-tubulin (Ganem et al., 2005), all constructs were expressed within a comparable range (±25% relative to the WT construct, except the ΔC construct, which was overexpressed by 64%; Fig. S2 A). We then determined the localization of the different constructs after RNAi-mediated depletion of both CLASP1 and CLASP2 (Fig. S2 B) to avoid functional redundancy (Mimori-Kiyosue et al., 2006; Pereira et al., 2006) and potential association with the endogenous proteins. We found that the 2ea-3eeaa construct failed to associate with mitotic spindle MTs (Fig. S3), whereas the IP12 construct failed to localize at growing MT plus-ends, as inferred by colocalization with EB1 in interphase cells (Fig. S4). In agreement, the 2ea-IP12 and IP12-3eeaa constructs showed a compromised localization at both spindle MTs and MT plus-ends (Figs. S3 and S4). To investigate the localization of the different CLASP2 constructs at unattached KTs we used mitotic cells treated with Nocodazole to depolymerize spindle MTs. All constructs, except ΔC and ΔC-Gcn4, were able to localize at unattached KTs (Fig. 2 B). Next, we quantified the fluorescence intensity of each CLASP2 construct relative to the signal from constitutive centromere proteins using anti-centromere antibodies (ACAs). As expected, mutant CLASP2 ΔC and CLASP2 ΔC-Gcn4 proteins were virtually undetectable at

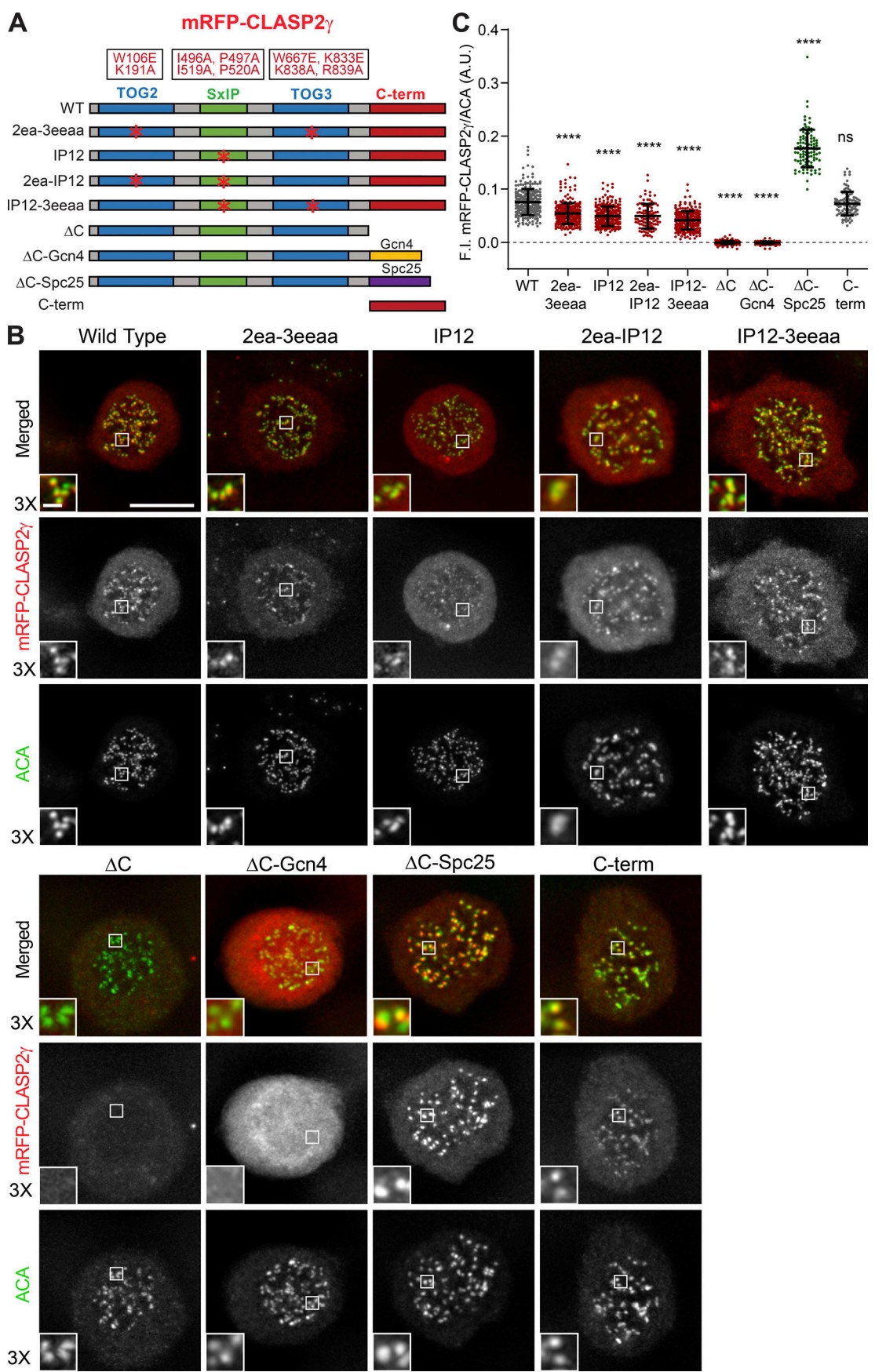

**Figure 2. CLASP2 C-term, but not self-association, is required and sufficient for KT localization. (A)** Schematic representation of the RNAi-resistant mRFP-CLASP2γ constructs. Mutations in TOG2 and TOG3 domains and SxIP motifs are indicated. **(B)** Localization of the different CLASP2 constructs at unattached KTs (+Nocodazole) upon endogenous CLASP depletion by RNAi. U2OS cells were immunostained with ACAs (green), and mRFP signal (red) was acquired directly. Magnified images of mRFP and ACA colocalization are shown. Scale bar is 5 µm. Scale bar in inset is 0.5 µm. **(C)** Quantification of KT fluorescence intensity (F.I.) of mRFP/ACA signals for each cell line. Each data point represents an individual KT. Bars represent mean and SD. Quantifications from a pool of three independent experiments. WT = 26 cells, 260 KTs; 2ea-3eeaa = 27 cells, 267 KTs; IP12 = 27 cells, 269 KTs; 2ea-IP12 = 10 cells, 101 KTs; IP12-3eeaa = 26 cells, 253 KTs; ΔC = 26 cells, 253 KTs; ΔC-Gcn4 = 18 cells, 196 KTs; ΔC-Spc25 = 10 cells, 100 KTs; C-term = 10 cells, 100 KTs. ****, P < 0.0001; t test.

unattached KTs, while CLASP2 C-term was sufficient to ensure normal KT accumulation (Fig. 2 C), confirming that the C-term of human CLASP2 is required and sufficient for KT localization, while demonstrating that dimerization itself is not sufficient to target human CLASP2 to KTs. This reveals a critical difference between human CLASP2 and its orthologue Stu1 in *S. cerevisiae*, in which dimerization was shown to be sufficient for KT targeting (Funk et al., 2014). Surprisingly, although all other CLASP2 mutants that compromise association with curved MT protofilaments and/or MT plus-end tracking were detectable at unattached KTs, they all showed a statistically significant decrease (~20%) relative to WT CLASP2 (Fig. 2 C). We attribute this reduction to the observed variability in the expression of the distinct constructs and/or possible alterations in the tridimensional conformation/folding of CLASP2 that might compromise its normal interaction with KTs. Lastly, the ΔC-Spc25 mutant showed an approximate twofold increase at unattached KTs relative to WT CLASP2 (Fig. 2 C). This likely reflects the constitutive KT localization of Spc25 and/or altered stoichiometry relative to endogenous CLASP2.

### Regulation of mitotic spindle length by CLASP2 depends on its KT localization and MT-binding properties

CLASPs have been previously implicated in the control of mitotic spindle length through their role (direct or indirect) in the incorporation of tubulin required for KT–MT poleward flux (Logarinho et al., 2012; Maffini et al., 2009; Maiato et al., 2003a; Maiato et al., 2005; Maiato et al., 2002; Mimori-Kiyosue et al., 2006), but the underlying mechanism remains unknown. One possibility is that, similar to members of the XMAP215/ch-TOG family that localize to KTs (Ayaz et al., 2012; Geyer et al., 2018; Miller et al., 2016; Nithianantham et al., 2018), CLASPs act as MT polymerases by interacting directly with soluble αβ-tubulin heterodimers. However, this hypothesis is not supported by our biochemical data. Alternatively, CLASPs might regulate spindle MT flux by controlling the interaction between KTs and the protofilaments of attached MT plus-ends, which were shown to have a curved configuration during both polymerization and depolymerization (McIntosh et al., 2008; McIntosh et al., 2018; McIntosh et al., 2013). To distinguish between these possibilities, we started by using fluorescence microscopy in fixed cells and the same RNAi rescue strategy to investigate how CLASP2 ensures normal mitotic spindle length. As expected, RNAi-mediated depletion of endogenous CLASPs from U2OS cells resulted in shorter spindles (Maffini et al., 2009), and expression of RNAi-resistant WT CLASP2γ fully rescued normal spindle length (Fig. 3, A and B). This further demonstrates the functional redundancy between CLASPs, while revealing that the TOG1

domain, which could potentially bind to free tubulin (Yu et al., 2016), is completely dispensable for the regulation of mitotic spindle length. In contrast, expression of all the other mutant constructs failed to rescue normal spindle length (Fig. 3, A and B). Interestingly, disruption of the TOG2 and TOG3 domains affected spindle length to a similar extent as the disruption of the SxIP motifs, whereas the combined disruption of both TOG2 or TOG3 domains and SxIP motifs exacerbated this effect, comparable with the disruption of the KT-targeting domain, expression of the KT-targeting domain only, or CLASP depletion (Fig. 3, A and B). These results indicate that recognition of growing MT plus-ends through EB–protein interaction and the ability to associate with curved MT protofilaments through TOG2 and TOG3 domains (but not TOG1-mediated binding to free αβ-tubulin heterodimers) are independently required to ensure normal spindle length. However, these MT-binding properties are insufficient to ensure the normal function of CLASP2, which requires targeting to KTs, independently of self-association. In agreement, driving monomeric CLASP2 to KTs through fusion with Spc25 partially rescued spindle length in the absence of CLASPs (Fig. 3, A and B).

### Chromosome congression, timely SAC satisfaction, and chromosome segregation fidelity rely on both CLASP2 KT- and MT-binding properties

Fixed-cell analysis revealed that expression of all mutant CLASP2 constructs resulted in a significant increase in anaphase and telophase cells with chromosome missegregation events (Fig. 4, A and B), suggesting a role for CLASP2 in error correction. To substantiate this, we resorted to live-cell imaging analysis by spinning-disk confocal microscopy, monitored chromosome movements after transduction of H2B-GFP in each of the stable U2OS lines expressing the distinct CLASP2 constructs, and added SiR-tubulin at 20 nM to visualize MTs (Lukinavičius et al., 2014). As expected, CLASP depletion resulted in a significant delay from nuclear envelope breakdown (NEB) to anaphase onset relative to controls (Fig. 5, A and B; and Video 1), and this delay was fully rescued by expression of the WT CLASP2γ construct, but not by any of the CLASP2 mutant constructs (Fig. 5, A and B; and Videos 2, 3, 4, 5, 6, 7, 8, 9, and 10). These results suggest that CLASP2 at the KT relies on its capacity to recognize growing MT plus-ends through EB–protein interaction and its ability to associate with curved MT protofilaments through TOG2 and TOG3 domains (but not TOG1-mediated binding to free αβ-tubulin heterodimers) to establish a functional KT–MT interface necessary for timely SAC satisfaction.

Of note, the observed delay in anaphase onset after CLASP depletion, or inefficient rescue with the different CLASP2 mutant constructs, was essentially due to a delay in efficiently

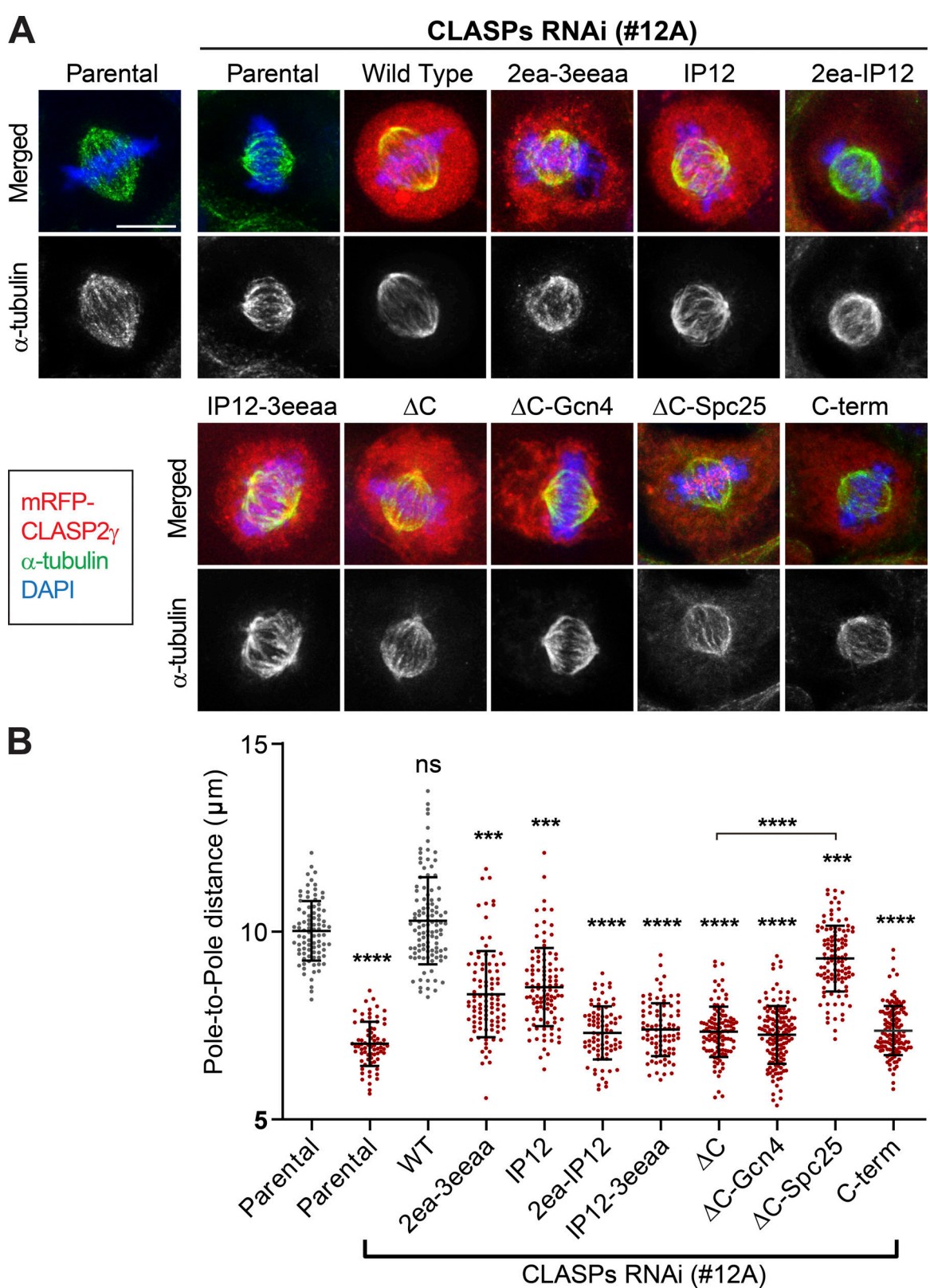

**Figure 3.** **Regulation of mitotic spindle length by CLASP2 depends on its KT localization and MT-binding properties. (A)** Immunofluorescence analysis of spindle length in the different U2OS PA–GFP–α-tubulin cell lines in metaphase. Scale bar is 5 µm. **(B)** Quantification of mitotic spindle length in each cell line. The first column represents the parental U2OS cell line and the remaining columns represent treatment with siRNA #12A. Each data point represents an individual cell. Bars represent mean and SD. Quantifications from a pool of three independent experiments: Parental = 88 cells; Parental with RNAi = 74 cells; WT = 108 cells; 2ea-3eeaa = 96 cells; IP12 = 105 cells; 2ea-IP12 = 82 cells; IP12-3eeaa = 88 cells; ΔC = 112 cells; ΔC-Gcn4 = 142 cells; ΔC-Spc25 = 113 cells; C-term = 123 cells. ***, P < 0.001; ****, P < 0.0001; *t* test.

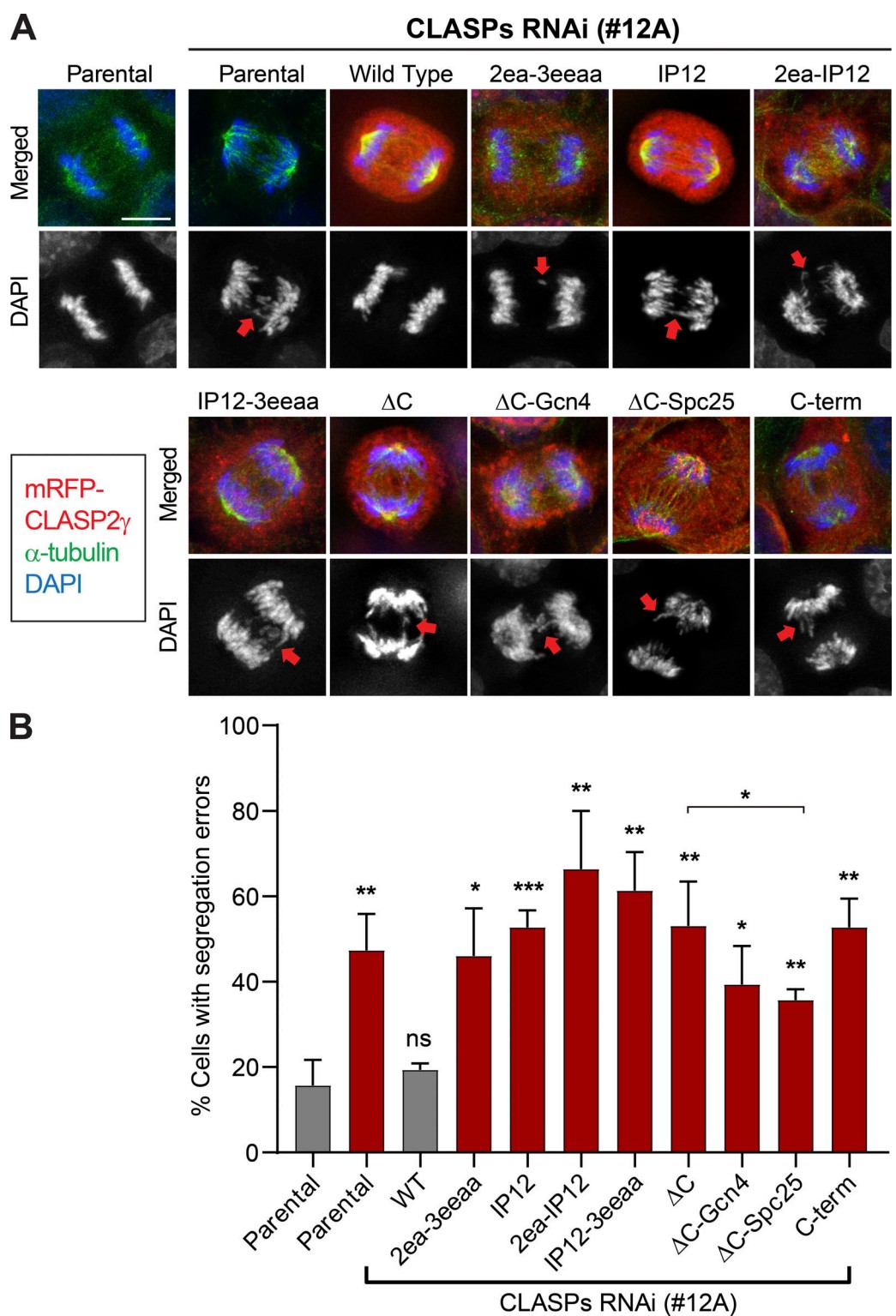

Figure 4. **CLASP2 KT localization and MT-binding properties are essential for chromosome segregation fidelity. (A)** Immunofluorescence analysis of the different U2OS PA–GFP–α-tubulin cell lines in anaphase. Segregation errors are indicated with red arrows in the DAPI channel. Parental and CLASP2 mutant cell lines depleted of endogenous CLASPs showed increased chromosome segregation errors relative to WT. Scale bar is 5 μm. **(B)** Quantification of chromosome segregation errors in anaphase and telophase cells. The first column represents parental U2OS cells without RNAi treatment; the remaining columns represent treatment with the siRNA #12A. Quantifications from a pool of three independent experiments: Parental = 92 cells; Parental with RNAi = 117 cells; WT = 153 cells; 2ea-3eeaa = 126 cells; IP12 = 135 cells; 2ea-IP12 = 53 cells; IP12-3eeaa = 121 cells; ΔC = 120 cells; ΔC-Gcn4 = 106 cells; ΔC-Spc25 = 86 cells; C-term = 73 cells. *, P < 0.05; **, P < 0.01; ***, P < 0.001; *t* test.

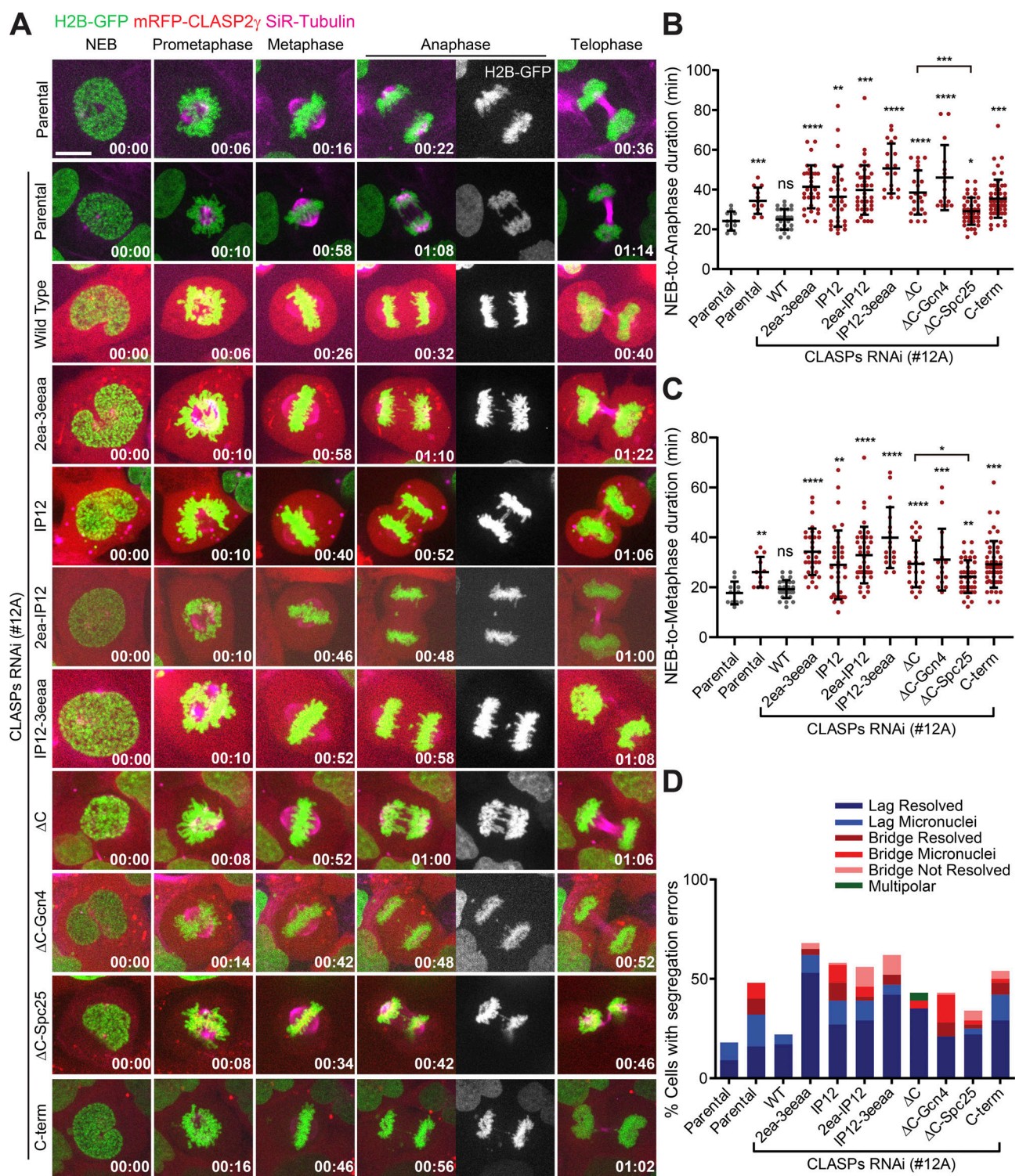

**Figure 5. Chromosome congression, timely SAC satisfaction, and chromosome segregation fidelity relies on both CLASP2 KT- and MT-binding properties. (A)** Live-cell imaging of U2OS PA–GFP–α-tubulin cells (without and with CLASPs RNAi) and expressing the different mRFP-CLASP2γ (red) constructs. DNA (blue) and SiR-tubulin (magenta) are also depicted. NEB = time 00:00. The anaphase panel also shows the H2B-GFP signal alone to highlight the segregation errors. Time is in hours:minutes. Scale bar is 5 µm. **(B)** Quantification of NEB-to-anaphase onset duration. First column represents parental U2OS PA–GFP–α-tubulin cells. The remaining columns represent CLASPs RNAi. Each data point represents an individual cell. Bars represent mean and SD. Parental without RNAi = 11 cells, pool of three independent experiments; Parental with RNAi = 12 cells, pool of three independent experiments; WT = 28 cells, pool of six independent experiments; 2ea-3eeaa = 32 cells, pool of 14 independent experiments; IP12 = 33 cells, pool of 13 independent experiments; 2ea-IP12 = 39 cells, pool of 10 independent experiments; IP12-3eeaa = 19 cells, pool of seven independent experiments; ΔC = 23 cells, pool of nine independent experiments; ΔC-Gcn4 = 15 cells, pool of five independent experiments; ΔC-Spc25 = 41 cells, pool of eight independent experiments; C-term = 52 cells, pool of seven independent experiments. *, P < 0.05; **, P < 0.01; ***, P < 0.001; ****, P < 0.0001; t test. **(C)** Quantification of the NEB-to-metaphase duration for the same dataset as in B. **(D)** Quantification of chromosome segregation errors for the same dataset as in B and C. Discrimination of the rates for each error type is represented.

completing chromosome congression to the spindle equator (Fig. 5, A and C; and Videos 1, 2, 3, 4, 5, 6, 7, 8, 9, and 10), consistent with a role of CLASP2 in the regulation of KT–MT attachments. Importantly, this delay correlated with an increase in chromosome segregation errors (mostly lagging chromosomes and some chromosome bridges) during anaphase and telophase (Fig. 5 D and Videos 1, 2, 3, 4, 5, 6, 7, 8, 9, and 10), in agreement with our fixed cell analysis (Fig. 4 B). Tracking the fate of these chromosome segregation errors revealed that while most lagging chromosomes and bridges resolved and reintegrated the main nuclei, a small fraction of the missegregated chromosomes resulted in the formation of micronuclei (Fig. 5 D), which have been associated with chromosome rearrangements commonly observed in human cancers (Crasta et al., 2012). Taken together, these results strongly suggest that multiple domains of CLASP2 ensure normal KT–MT attachments required for chromosome congression, timely SAC satisfaction and efficient error correction during mitosis.

## CLASP2 relies on both KT localization and its MT-binding properties for the establishment of robust KT–MT attachments

To directly investigate whether and how the multiple CLASP2 domains contribute to establish functional KT–MT attachments, we first assessed the formation of robust KT-fibers (K-fibers) by subjecting metaphase-like cells (treated with 5 µM MG132 for 1 h) depleted of endogenous CLASPs and expressing the different CLASP2 constructs to a short cold shock. As a positive control, we compared the effects with metaphase-like cells depleted of the outer KT component Ndc80/Hec1, previously implicated in the establishment of end-on KT–MT attachments (McCleland et al., 2003; McCleland et al., 2004; Wigge and Kilmartin, 2001). As expected, K-fiber formation was strongly compromised in Ndc80-depleted metaphase-like cells, as confirmed by ~70% reduction in MT signal intensity at KTs (Fig. 6, A–C). In contrast, cold-stable K-fibers were clearly identifiable after endogenous CLASP depletion and in all CLASP2 WT and mutant rescue experiments, but the signal intensity was reduced up to 50% in the different MT- and KT-binding mutants relative to our negative control RNAi in parental U2OS cells (Fig. 6, A–C). This suggests that CLASP2 relies on both KT localization and its MT-binding properties for the establishment of robust KT–MT attachments. Surprisingly, depletion of endogenous CLASPs showed even a slight increase in K-fiber signal relative to the negative control in this assay, despite efficient (~80%) depletion by RNAi (Fig. 6, A–C; and Fig. S2 B). One possibility is that residual CLASP2 protein after RNAi depletion is sufficient to ensure the formation of robust K-fibers in this assay. In this case, the expression of the CLASP2 mutant constructs could cause a dominant negative effect, which, together with the partial depletion of endogenous CLASPs, could lead to K-fiber destabilization. Alternatively, this fixed-cell analysis might compromise the distinction between CLASP-depleted late prometaphase and metaphase cells (despite MG132 treatment), which would be consistent with our previous measurements of KT–MT half-life after CLASP depletion in a mixed prometaphase/metaphase population, which suggested a role for CLASPs in KT–MT destabilization (Maffini et al., 2009). Importantly, the ΔC-Spc25

construct partially recovered K-fiber robustness relative to the ΔC mutant, suggesting that CLASP2 can function as a monomer at the KT (note that the monomeric character of CLASP2 in the chimeric fusion with Spc25 is irrespective of the ability of Spc25 to heterodimerize with Spc24 in the context of the Ndc80 complex).

The inability of cells expressing the different CLASP2 mutants to form robust K-fibers is consistent with the observed delay in satisfying the SAC. To directly investigate this we used the same experimental setup described above to infer K-fiber formation and quantified the frequency of metaphase-like cells with clear Mad1 accumulation at KTs as a proxy for incomplete MT occupancy (Kuhn and Dumont, 2019). While the frequency of metaphase-like cells with Mad1-positive KTs was, as expected, strongly increased in Ndc80-depleted cells, this was only slightly increased (without statistical significance due to high cell-to-cell variability) in very few CLASP-depleted cells, with or without the expression of the WT and mutant constructs (Fig. 6, D and E). Altogether, these results are consistent with a role for CLASP2 KT- and MT-binding properties in the formation of robust K-fibers, which, despite a delay, ultimately satisfy the SAC and allow the completion of chromosome congression.

## CLASP2-mediated recognition of growing MT plus-ends and the ability to associate with curved MT protofilaments promotes flux and stabilizes KT attachments on bi-oriented chromosomes

To directly test how the different CLASP2 domains impact KT–MT dynamics, we quantified KT- and non-KT–MT half-life by fluorescence dissipation after photoactivation (Girão and Maiato, In press). By fitting the fluorescence decay over time to a double exponential curve, it is possible to discriminate two spindle MT populations with fast and slow turnover, which are thought to correspond to the less stable non-KT–MTs and more stable KT–MTs, respectively (Bakhoum et al., 2009a; Bakhoum et al., 2009b; Zhai et al., 1995). Admittedly, while this approach might be unable to distinguish multiple stable MT populations that are likely to exist in the spindle (Tipton and Gorbsky, 2019 *Preprint*), it remains the gold standard to detect subtle alterations in KT–MT dynamics that cannot be disclosed by less sensitive methods, such as cold shock. In addition, by measuring the poleward movement of the photoactivated fluorescent mark in the spindle relative to the metaphase plate (inferred by a fluorescence exclusion zone in the region of the chromosomes), we determined the respective poleward flux rates. We found that CLASP depletion resulted in a significant decrease in KT–MT half-life and poleward flux during metaphase when compared with controls (Fig. 7, A, B, and D; and Fig. S5, A and B) without significantly affecting the half-life of non-KT–MTs, nor the relative distribution of KT- and non-KT–MTs (Fig. 7, A and C; and Fig. S5, A–D). Importantly, both WT CLASP2γ and ΔC-Spc25 constructs were able to fully rescue normal KT–MT half-life and poleward flux (Fig. 7, A, B, and D; and Fig. S5, A and B), indicating dispensability of the TOG1 domain and the ability to self-associate at KTs for poleward flux and the stabilization of attached MTs during metaphase. In contrast, mutation of the TOG2 and TOG3 domains or the SxIP motifs of CLASP2 resulted

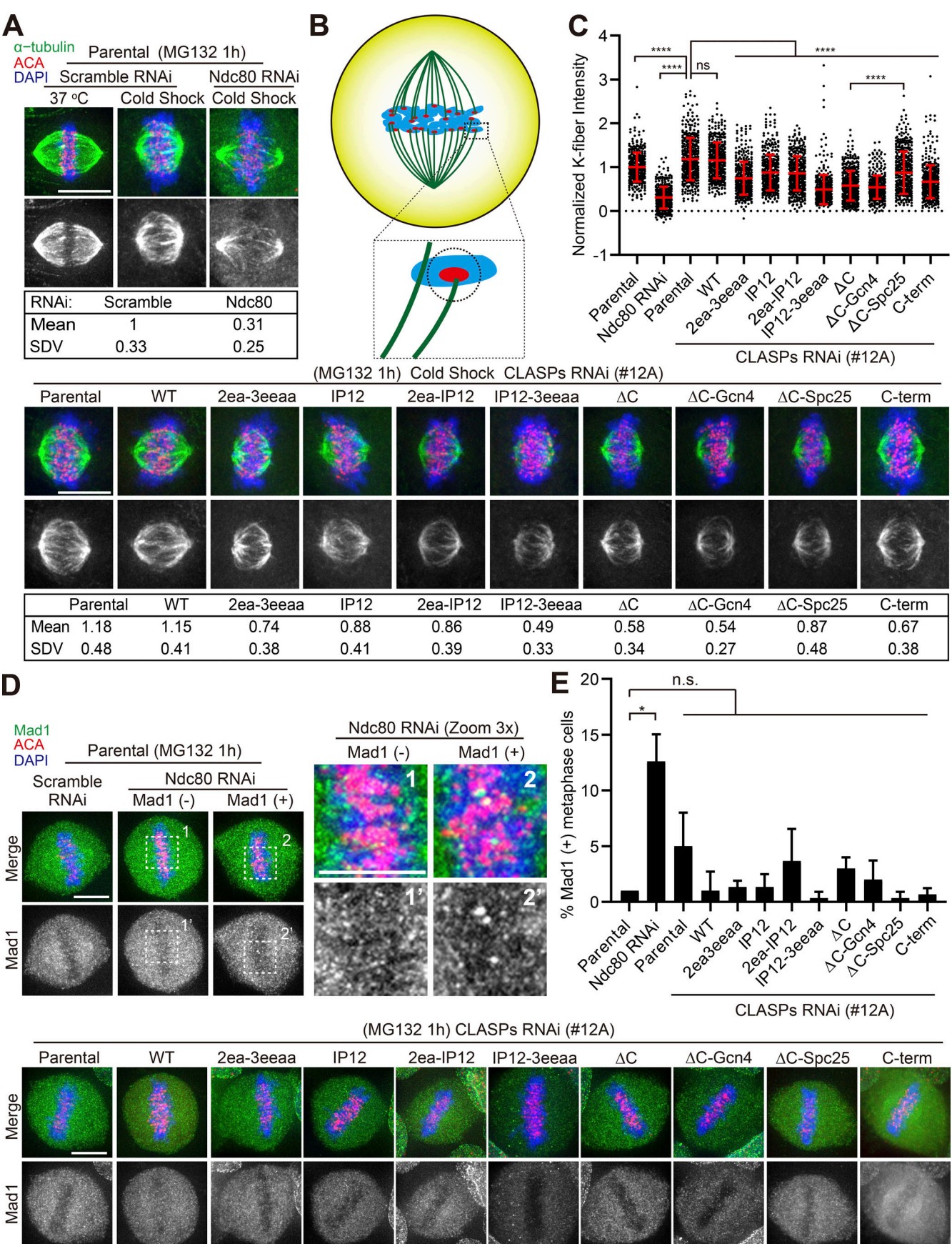

Figure 6. **CLASP2 is required for the formation of robust K-fibers that ultimately satisfy the SAC. (A)** Metaphase-arrested U2OS parental cells and mRFP-CLASPγ–expressing cell lines after CLASP RNAi were fixed and stained after a short cold shock. Ndc80 RNAi was used as a positive control. The respective mean and standard deviation (SDV) of the K-fiber signal intensity are indicated for all experimental conditions. Scale bar is 10 µm. **(B)** Graphical representation of the normalized K-fiber signal quantification. The corresponding signal of α-tubulin in the close vicinity of each KT was quantified and background subtracted (highlighted, black circle). **(C)** Quantification of K-fiber signal intensity. Each signal was normalized to the mean signal of Scramble siRNA in parental cells (bars represent mean and SD. Parental scramble = 389 KTs; siNdc80 = 407 KTs; Parental with CLASPs RNAi = 396 KTs; WT = 390 KTs; 2ea-3eeaa = 395 KTs; IP12 = 408 KTs; 2ea-IP12 = 415 KTs; IP12-3eeaa = 424 KTs; ΔC = 418KTs; ΔC-Gcn4 = 421 KTs; ΔC-Spc25 = 408 KTs; C-term = 427 KTs. n = 21 cells per condition, pool of three independent experiments. ****, P < 0.0001; two-tailed Welch's t test. **(D)** Metaphase-arrested U2OS parental cells and mRFP-CLASP2γ–expressing cell lines after CLASPs RNAi were fixed and stained in order to quantify the percentage of cells with Mad1-positive KTs. The insets in the top panel represent higher magnification of selected regions. Scale bar is 10 µm. **(E)** Quantification of the percentage of cells with Mad1-positive KTs for each cell line. Bars represent mean and SD. n = 300 cells for each cell line from a pool of three independent experiments, with the exception of siNdc80 (n = 232). *, P < 0.05; two-tailed Welch's t test.

in a flux reduction and less stable KT–MTs with the consequent inability to rescue the effect caused by CLASP depletion, a tendency that was exacerbated by the combined mutation of the TOG2 or TOG3 with the SxIP domains (Fig. 7, A, B, and D; and Fig. S5, A and B). Of note, although all the ΔC, as well as the C-term and 2ea-IP12 mutants, did show statistically significant differences in the distribution of KT– and non-KT–MTs (Fig. S5, C and D), the variability relative to parental controls was <15%, clearly demonstrating that both populations are present in all experimental conditions in agreement with our cold-shock data (Fig. 6, A–C). These results suggest that the ability of CLASP2 to recognize growing MT plus-ends and associate with curved MT protofilaments act synergistically to promote flux and stabilize KT–MT attachments during metaphase in human cells.

The inability to rescue normal flux rates and KT–MT half-life after CLASP depletion was maximal in the C-terminal deletion mutants ΔC and ΔC-Gcn4, as well as in the CLASP2 mutant containing only the KT-binding domain (C-term; Fig. 7, A, B, and D; and Fig. S5, A and B). This indicates that although the localization of CLASP2 at KTs is not sufficient to ensure normal poleward flux and KT–MT stability, it is critical for normal KT–MT dynamics. Remarkably, with the noticeable exception of the 2ea-IP12 CLASP2 mutant, which caused a 25% decrease in non-KT–MT half-life for reasons we do not fully understand, no other mutant significantly compromised non-KT–MT half-life within the temporal resolution limit of our assay (Fig. 7, A and C; and Fig. S5, A and B). We conclude that the role of CLASP2 in the regulation of mitotic spindle length, chromosome congression, error correction, and timely SAC satisfaction relies essentially on its ability to integrate critical MT-binding properties at the KT–MT interface that sustains MT growth required for poleward flux and regulate KT–MT attachment stability.

## Discussion

Through systematic functional dissection of structure-guided CLASP2 mutants compromised in distinct MT- and KT-binding properties, complemented with in-depth biochemical analyses of the CLASP2 protein, our results support a model in which monomeric CLASP2, targeted to KTs through its C-term, integrates multiple independent features, including the recognition of growing MT plus-ends through EB–protein interaction and the ability to associate with curved MT protofilaments through TOG2 and TOG3 domains, to promote growth and

stabilization of attached MTs required for poleward flux (Fig. 8). This model is consistent with the unique ability of human CLASPs to bind to curved protofilaments at MT plus-ends (Leano et al., 2013; Leano and Slep, 2019; Maki et al., 2015), in striking contrast with members of the XMAP215/ch-TOG family, which work as MT polymerases that are able to bind to free αβ-tubulin heterodimers (Ayaz et al., 2012; Geyer et al., 2018; Nithianantham et al., 2018; Yu et al., 2016). Moreover, this model is supported by recent EM and 3D electron tomography work showing that both growing and shrinking MT plus-ends, including those of KT–MTs, are curved (McIntosh et al., 2008; McIntosh et al., 2018; McIntosh et al., 2013). Finally, CLASP1 (but not CLASP2, likely due to its relatively low abundance in human U2OS cells) was shown to have the capacity to bind to curled tubulin oligomers in mitotic cell lysates (Volkov et al., 2015). Given the strong similarity between equivalent TOG domains of human CLASP1 and CLASP2 (Leano et al., 2013; Leano and Slep, 2019; Maki et al., 2015) and the strong functional redundancy between both human CLASP proteins for mitosis (Mimori-Kiyosue et al., 2006; Pereira et al., 2006), it is reasonable to conceive that the findings reported here for CLASP2 can be extended to CLASP1. Importantly, neither the observed variability (∼20% relative to CLASP2 WT) in protein expression or accumulation at the KTs for the different CLASP2 mutant constructs accounts for the observed mitotic phenotypes, given that cell viability and essentially normal mitosis can be assured by expression of a single CLASP protein (Mimori-Kiyosue et al., 2006; Pereira et al., 2006).

Our findings strongly suggest that CLASP2-assisted poleward flux and stabilization of KT–MT attachments do not require binding to free αβ-tubulin heterodimers, which are proposed to be essentially mediated by the TOG1 domain (Yu et al., 2016). Interestingly, recent determination of the x-ray crystal structure of the TOG1 domain of human CLASP1 revealed that, while bearing structural similarities to the free tubulin-binding TOG domains of the XMAP215/ch-TOG family of MT polymerases, it lacks many key features required to do so (Leano and Slep, 2019). Thus, a picture in which KT CLASPs associate with curved protofilaments at MT plus-ends to prevent catastrophes and promote rescue upon, or in coordination with, XMAP215/ch-TOG family-mediated recruitment of new αβ-tubulin heterodimers, is emerging. The significance, if any, of CLASPs binding to αβ-tubulin heterodimers, as well as CLASPs capacity to self-associate, remains to be determined, namely in other cellular contexts outside mitosis.

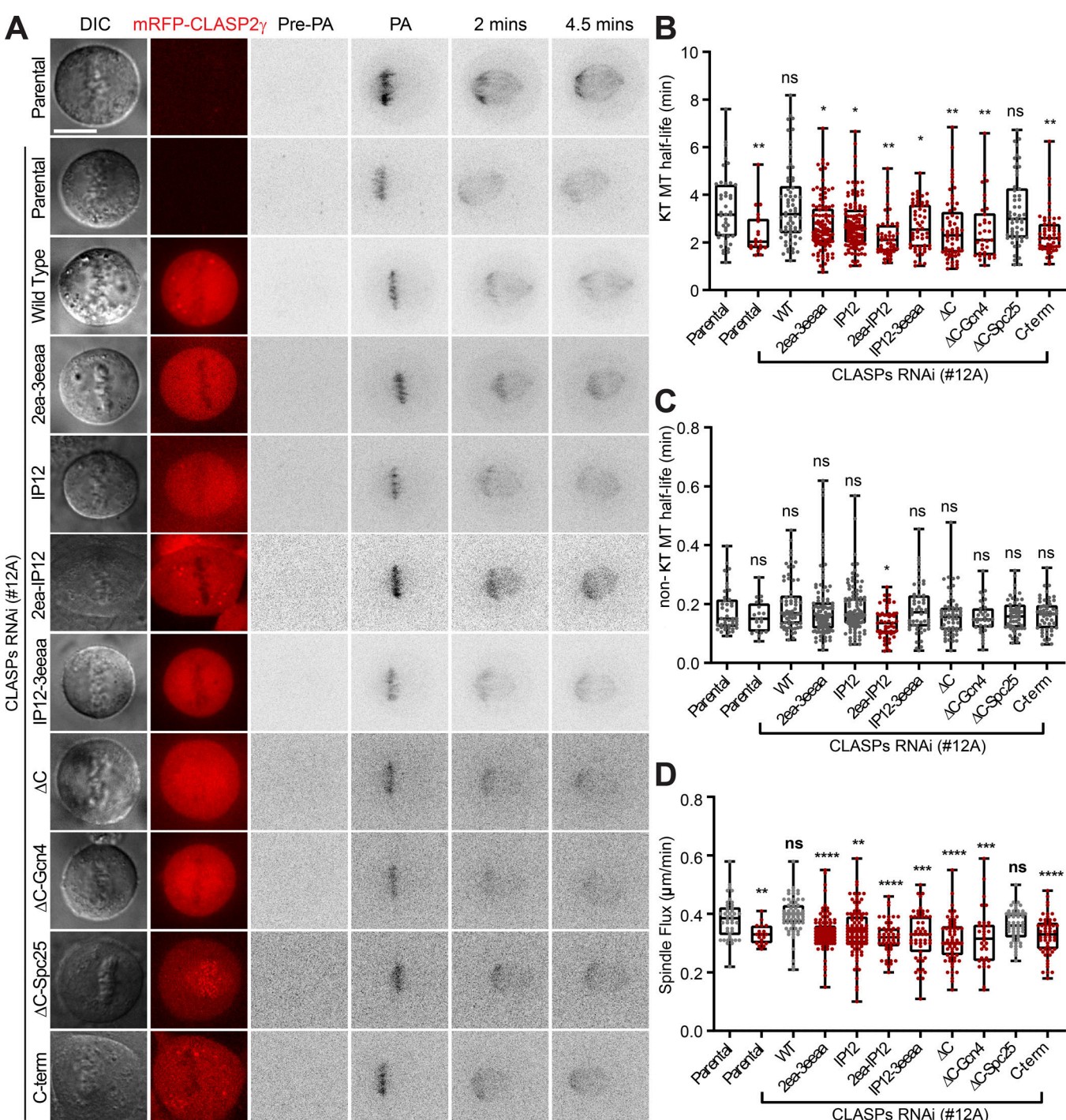

Figure 7. **Multimodal regulation of KT–MT dynamics by CLASP2. (A)** Photoactivation experiments in live U2OS parental PA–GFP–α-tubulin cells and expressing the different mRFP-CLASP2γ constructs after CLASP RNAi. Panels represent DIC, the mRFP-CLASP2γ signal (red), and the PA–GFP–α-tubulin signal before photoactivation (Pre-PA), immediately after photoactivation (PA), and at 2 and 4.5 min after photoactivation. The PA–GFP–α-tubulin signal was inverted for better visualization. Scale bar is 5 μm. **(B)** Quantification of KT–MT half-life for all cell lines. The first column corresponds to the parental cell line and the remaining columns correspond to experiments performed after CLASP depletion. Each data point represents an individual cell. The boxes represent median and interquartile interval; the bars represent minimum and maximum values. Parental = 43 cells, pool of five independent experiments; Parental with RNAi = 20 cells, pool of six independent experiments; WT = 66 cells, pool of seven independent experiments; 2ea-3eeaa = 105 cells, pool of eight independent experiments; IP12 = 108 cells, pool of nine independent experiments; 2ea-IP12 = 52 cells, pool of 11 independent experiments; IP12-3eeaa = 55 cells, pool of four independent experiments; ΔC = 60 cells, pool of four independent experiments; ΔC-Gcn4 = 34 cells, pool of eight independent experiments; ΔC-Spc25 = 54 cells, pool of 12 independent experiments; C-term = 53 cells, pool of 12 independent experiments. *, $P < 0.05$; **, $P < 0.01$; ***, $P < 0.001$; ****, $P < 0.0001$; Mann-Whitney U-Test. **(C)** Quantification of non-KT–MT half-life for all cell lines for the same dataset as in B. **(D)** Quantification of spindle poleward flux for all cell lines and same dataset as in B and C.

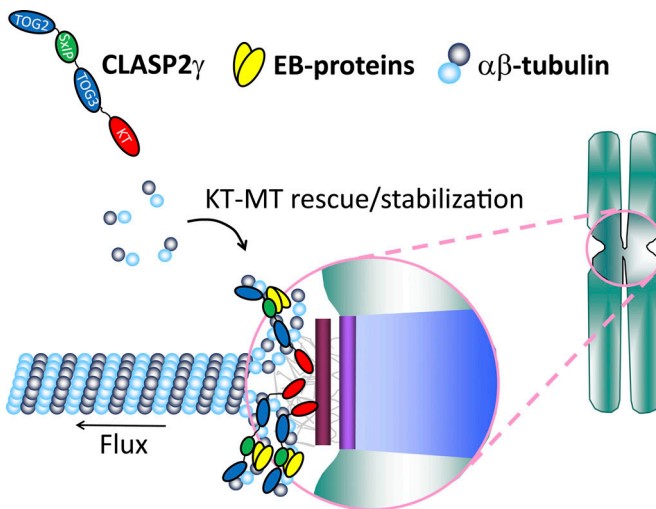

**CLASP2γ** **EB-proteins** **αβ-tubulin**

KT–MT rescue/stabilization

Flux

Figure 8. **Proposed model for CLASP2-mediated stabilization of KT–MT attachments and promotion of poleward flux.** Monomeric CLASP2, targeted to KTs through its C-term (red), integrates multiple independent features, including the recognition of growing MT plus-ends through EB–protein (yellow) interaction and the potential to associate with curved MT protofilaments through TOG2 and TOG3 domains (blue) to promote growth and stabilization of attached MTs required for poleward flux (see main text for details).

While the curved conformation of polymerizing and depolymerizing KT-attached MT plus-ends could not be distinguished at the ultrastructural level (McIntosh et al., 2008; McIntosh et al., 2018; McIntosh et al., 2013), our study indicates that CLASPs might be able to recognize these two functional states through interaction with EB–proteins that specifically track growing MT plus-ends and were recently shown to distinguish their structural state (Reid et al., 2019). Interestingly, although not fully abolished, CLASP2 association with EB1 at growing MT plus-ends is negatively regulated by CDK1- and GSK3β-mediated phosphorylation close to the CLASP2 SxIP motifs during mitosis (Kumar et al., 2012; Kumar et al., 2009; Wittmann and Waterman-Storer, 2005), and this fine regulation might be important for chromosome segregation fidelity (Pemble et al., 2017). Indeed, our findings revealed that mutation of the CLASP2 SxIP motifs was sufficient to compromise efficient error correction, giving rise to chromosome segregation errors. In this regard, it is difficult to conceive that the chromosome segregation errors observed, not only in the IP12 mutant, but also in all other mutants evaluated in the present study, result from less stable MT attachments during metaphase, as our measurements of KT–MT half-life seem to suggest. Most likely, these chromosome segregation errors resulted from a compromised MT-destabilizing role of CLASP2 earlier in mitosis, in agreement with previous measurements of KT–MT half-life after human CLASP depletion, in which late prometaphase and metaphase cells were not distinguished (Maffini et al., 2009). This is consistent with the idea that CLASPs are part of a CDK1-dependent regulatory switch that controls the transition between labile-to-stable KT–MT attachments (Maia et al., 2012; Manning et al., 2010) and previous findings indicating that KT–MT turnover is faster in

prometaphase (Kabeche and Compton, 2013). Importantly, despite possible weak interactions between the TOG2 (but not TOG3) domain of CLASP2 with depolymerizing MTs (Maki et al., 2015), recent in vitro reconstitution experiments showed that CLASP2- and CENP-E-coated spherical beads were able to autonomously mediate moderately long-lived attachments with polymerizing, but not depolymerizing, MT plus-ends (Chakraborty et al., 2019). The ability of TOG3 domains of CLASP2 to bridge neighboring MT protofilaments (Maki et al., 2015; Leano and Slep, 2019) might be instrumental to rescue KT–MT depolymerization, thereby stabilizing KT–MT attachments and promoting poleward flux on bi-oriented chromosomes (Fig. 8).

## Materials and methods

### Production of recombinant full-length CLASP2 protein
Human CLASP2 FL (UniProt accession no. O75122-3) was purified using Baculovirus expression vector system (BD Biosciences). A synthetic gene coding for the human CLASP2 FL (including amino acids Met1–Ser1515) with codon usage optimized for expression in insect cells, was obtained from GenScript. CLASP2 ORF was subcloned into the *BamHI* and *XbaI* restriction sites of the expression vector pKL (Addgene, #110741), in fusion with an N-terminal Strep-tag (MASWSHPQFEKSGGGGGENLYFQG) and a C-terminal His10-tag, yielding plasmid pKL-CLASP2-FL. Integration of pKL-CLASP2-FL into the MultiBac baculoviral genome was performed by Tn7 transposition using DH10EMBacY *Escherichia coli* cells. Production of recombinant baculovirus and transfection of *Spodoptera frugiperda* Sf21 cells was performed using the MultiBac expression system (Bieniossek et al., 2008). Sf21 cells were lysed by sonication in 50 mM Tris-HCl, pH 8.0, 150 mM NaCl, and 7 mM β-mercaptoethanol supplemented with protease inhibitors (Complete EDTA-free, Roche). Clarified protein extracts were loaded onto a HisTrap HP column (GE Healthcare) preequilibrated in 50 mM Tris-HCl, pH 8.0, 500 mM NaCl, 20 mM imidazole, and 7 mM β-mercaptoethanol and eluted with 200 mM imidazole. CLASP2 FL–containing fractions were pooled, adjusted to 100 mM Tris-HCl, pH 8.0 and 1 mM EDTA, and loaded onto a StrepTrap HP column (GE Healthcare) preequilibrated in 100 mM Tris-HCl, pH 8.0, 150 mM NaCl, 1 mM EDTA, and 1 mM DTT. Bound CLASP2 was eluted with 2.5 mM desthiobiotin. Protein-containing fractions were concentrated and further purified on a HiPrep 16/60 Sephacryl S-300 HR column (GE Healthcare) preequilibrated with 20 mM Hepes, pH 7.0, 400 mM NaCl, 5% glycerol, and 1 mM β-mercaptoethanol. Pure protein was concentrated using a 30-kD cutoff ultracentrifugal concentration device (Millipore).

### Production of recombinant C-terminal CLASP2 proteins
A synthetic gene coding for the C-terminal comprising amino acids Asp1213–Ser1515 of human CLASP2α with codon usage optimized for expression in *E. coli* was obtained from GenScript. CLASP2 ORF was subcloned into the *EcoRI* and *NdeI* restriction sites of the expression vector pPR-IBA2 (IBA Lifesciences), in fusion with an N-terminal Strep-tag and a C-terminal His10-tag (pPR-IBA2-Strep-CLASP2-C-term-His10). Site-directed mutagenesis of this vector was made to add a codon stop before the

C-terminal His10-tag (pPR-IBA2-Strep-CLASP2-C-term). *E. coli* BL21 Star (DE3) cells (Life Technologies) transformed with pPR-IBA2-Strep-CLASP2-C-term-His10 and pPR-IBA2-Strep-CLASP2-C-term plasmids were grown at 37°C in Luria broth medium supplemented with 50 µg/ml ampicillin to OD600 0.4, and expression was induced by addition of 0.4 mM IPTG. After growing for 4 h at 37°C, cells were harvested and lysed by sonication in 50 mM Tris-HCl, 150 mM NaCl, pH 8.0, and supplemented with protease inhibitors (Complete EDTA-free, Roche). Clarified protein extracts were loaded onto a HisTrap HP column (GE Healthcare) preequilibrated in 50 mM Tris-HCl, 500 mM NaCl, 20 mM imidazole, pH 8.0, and eluted with 200 mM imidazole. CLASP2 C-term or CLASP2 C-term without His-tag containing fractions were pooled, adjusted to 100 mM Tris-HCl, pH 8.0, and 1 mM EDTA, and loaded onto a StrepTrap HP column (GE Healthcare) preequilibrated in 100 mM Tris-HCl, pH 8.0, 150 mM NaCl, and 1 mM EDTA. Bound CLASP2 was eluted in the buffer supplemented with 2.5 mM desthiobiotin. Protein-containing fractions were concentrated and further purified on a HiPrep 16/60 Sephacryl S-200 HR column (GE Healthcare) preequilibrated with 50 mM Tris-HCl, 400 mM NaCl, 5% glycerol, and 1 mM EDTA, pH 8.0. Pure protein was concentrated using a 10-kD cutoff ultracentrifugal concentration device (Millipore). For some experiments, the N-terminal Strep-tag of CLASP2 C-term was also removed by cleavage at a tobacco etch virus protease recognition site immediately after the tag (CLASP2 C-term without Strep-tag).

### HeLa protein extracts for SEC and sucrose gradient analysis
Protein extracts were obtained from Nocodazole-treated mitotic HeLa cells by addition of lysis buffer (20 mM Hepes/KOH, 400 mM NaCl, 0.1% Triton X-100, 1 mM DTT, 1.4 mM PMSF, and 1:200 protease inhibitor cocktail [Sigma-Aldrich, P8340], pH 7.0) and sonication in a Branson Sonifier 250 with a micro tip (3 × 3 s, 8% power output, and 40% duty cycle). The suspension was centrifuged for 20 min at 13,000 rpm at 4°C and the resulting supernatant was filtered through a 0.22-µm cellulose acetate filter. Aliquots were quickly frozen in $LN_2$ and stored at –80°C. Total protein levels were quantified by Bradford protein assay.

### SEC
SEC experiments with HeLa cell extracts or recombinant proteins (CLASP2 FL and CLASP2 C-term) were performed with a prepacked Superose 6 10/300 GL column (GE Healthcare) equilibrated with 20 mM Hepes, 400 mM NaCl, 5% glycerol, 1 mM EDTA, and 1 mM DTT, pH 7.0, at a flow rate of 0.4 ml/min, or with prepacked Superose 12 10/300 GL column (GE Healthcare) equilibrated with 50 mM Tris, 400 mM NaCl, 5% glycerol, and 1 mM EDTA, pH 8.0, at a flow rate of 0.5 ml/min. Calibration was performed with thyroglobulin (8.50 nm), ferritin (6.10 nm), aldolase (4.81 nm), ovalbumin (3.05 nm), and ribonuclease A (1.64 nm). 1 mg of cell extract and 100–300 µg of recombinant proteins were used in each run. For the HeLa extract runs, 1 ml fractions were collected, TCA precipitated, subjected to SDS-PAGE, blotted, and immunodetected with a rat anti-CLASP2 antibody (Maffini et al., 2009). Protein distribution was analyzed by densitometry. SEC analyses were also performed at different pH and ionic strength conditions (50 mM Tris-HCl, pH 8.0, and/or 150 mM NaCl) to exclude their possible effect in the determination of the native oligomeric states.

### Density gradient centrifugation
Continuous gradients of 5–20% sucrose were prepared in 20 mM Hepes, 400 mM NaCl, 1 mM EDTA, and 1 mM DTT, pH 7.0, using a Gradient Master (BioComp Instruments). 40–80 µg of protein standards (soybean trypsin inhibitor [2.3 S], albumin [4.3 S], IgG [6.9 S], and catalase [11.3 S]) together with either recombinant, CLASP2 FL (~40 µg) or CLASP2 C-term (~70 µg), or 1 mg of HeLa cell extract were loaded on top of the gradients and centrifuged at 38,000 rpm in a SW41 rotor (Beckman) for 26 h at 4°C. After centrifugation, 0.75-ml fractions were collected from the bottom of the tubes, TCA precipitated, and analyzed by SDS-PAGE. For the HeLa extract analysis, the gels were blotted onto a nitrocellulose membrane, Ponceau S–stained, destained, and then probed with anti-CLASP2 antibody. For the protein standards analysis, gels were simply stained with Coomassie brilliant blue. Protein sedimentation profiles were analyzed by densitometry. The sedimentation behavior of CLASP2 FL and CLASP2 C-term was also analyzed in 50 mM Tris-HCl, 150 mM NaCl, and 1 mM DTT, pH 8.0.

### Estimation of MW by SEC and density gradient
The MWs of endogenous and recombinant CLASP2 FL and recombinant CLASP2 C-term were calculated by the Siegel and Monty method (Siegel and Monty, 1966) using the $R_s$ and sedimentation coefficients obtained from SEC and density gradient centrifugations, respectively. The following simplified equation was used (Erickson, 2009): $MW = R_s \times s \times 4.205$, where $s$ is the sedimentation coefficient, and $R_s$ is the Stokes radius.

### Estimation of mutual Dm and hydrodynamic radius by DLS
The mutual Dm of recombinant CLASP2 C-term was determined at different concentrations ($c$; 1, 2, 4, 6, 8, and 10 mg/ml in 50 mM Tris-HCl, 400 mM NaCl, 5% glycerol, and 1 mM EDTA, pH 8.0) using a quartz cuvette ZEN2112 and a Zetasizer Nano ZS DLS system (Malvern Instruments). Three independent measurements were obtained at 25°C and the acquired correlogram (correlation function versus time) and Dm were analyzed using Zetasizer software v7.03. The representation of Dm ($\mu m^2\ s^{-1}$) as a function of the protein concentration (g/ml) allowed the determination of the self-Dm (Ds; value of Dm at Y interception of the plot, $c = 0$) and of the interaction parameter (slope of the plot/Ds; Yadav et al., 2011).

### ITC
The measurements of CLASP2 C-term heat of dissociation were performed at 25°C using a MicroCal VP-ITC calorimeter (GE Healthcare Life Sciences). All proteins were dialyzed against 2 liters of ITC buffer (50 mM Tris-HCl, 150 mM NaCl, 5% glycerol, and 1 mM EDTA, pH 8.0) using a 2-kD cutoff Slide-A-Lyzer dialysis cassette (Thermo Scientific). After dialysis, the proteins were centrifuged for 30 min at 13,200 rpm at 4°C to ensure the removal of putative protein aggregates. The calorimeter cell sample (1.4 ml) was loaded with ITC buffer and the titration

assay was performed by 34 sequential 8-µl injections of CLASP2 C-term (600 µM) with 5-min intervals. Controls included BSA (600 µM) and ITC buffer. All titration assays were performed twice. ITC raw data were analyzed using the NITPIC (for baseline selection and peak integration; Keller et al., 2012) and Origin 7.0 (Microcal) programs.

### Constructs and lentiviral transduction

The mRFP-CLASP2γ constructs WT, IP12 (I496A, P497A; and I519A, P520A), 2ea-3eaa (W106E, K191A; and W667E, K833E, K838A, R839A), 2ea-IP12 (W106E, K191A; I496A, P497A; and I519A, P520A), IP12-3eeaa (I496A, P497A; I519A, P520A; and W667E, K833E, K838A, R839A), and ΔC (CLASP2γ without C-term 1017–1294) inserted between NdeI-NotI sites in a CSII-CMV-MCS vector (RIKEN BRC DNA Bank catalog no. RDB04377) were reported previously (Grimaldi et al., 2014; Maki et al., 2015). Of note, sequence analysis revealed a two-nucleotide deletion in the C-terminal of the mRFP-CLASP2γ WT construct causing a frameshift at amino acid position 1267 (Lys-Asn). This caused the premature termination of the ORF by a stop codon at amino acid position 1285, which did not compromise CLASP2γ function, as shown by the RNAi-rescue experiments. The ΔC-Gcn4 construct was generated by inserting the Gcn4 sequence after the mRFP-CLASP2γ-ΔC sequence, using Gibson assembly (Gibson et al., 2009). The entire Gcn4 sequence was contained in the two primers used in the reaction mixed with the vector containing the mRFP-CLASP2γ-ΔC construct cut with NotI. To generate the ΔC-Spc25 construct, the Spc25 sequence was amplified from p3xHalo-GFP-Spc25 (gift from M. Lampson, University of Pennsylvania, Philadelphia, PA; Zhang et al., 2017) by PCR. The vector containing mRFP-CLASP2γ-ΔC construct was cut with BstB1 and NotI, and then mixed with the PCR fragment containing Spc25 using Gibson assembly. To generate the C-term construct, the vector containing mRFP-CLASP2γ-WT was cut with BstB1 and Mlu1. The resulting fragment containing the C-terminal sequence was mixed with the fragment containing the mRFP sequence obtained by PCR using Gibson assembly. Lentiviral supernatants were produced in HEK293T cells using pMD2.G (Addgene, #12259), psPAX2 (Addgene, #12260), and the CSII-CMV-MLS plasmids with the presence of the Lipofectamine 2000 (Invitrogen) and Opti-MEM (Gibco). A U2OS cell line stably expressing PA-GFP–α-tubulin (Ganem et al., 2005) was transduced with the viral supernatants in the presence of 6 µg/ml Polybrene (Sigma-Aldrich). The same procedure was used to produce the double expressing H2B-GFP and mRFP-CLASP2γ cell lines.

### Cell line maintenance

All cell lines used in this work were cultured in DMEM with 10% FBS, supplemented with 10 µg/µl of antibiotic-antimycotic mixture (Gibco) and selected with Zeocine 1 µg/µl. Cell lines were maintained at 37°C in a 5% $CO_2$, humidified atmosphere. To synchronize HeLa cell cultures in mitosis, 3.3 µM Nocodazole was added to the medium 16 h before harvesting by mitotic shake-off.

### Cell sorting

Cells expressing the different mRFP-CLASP2γ constructs were enriched by cell sorting using a FACS Aria II cell sorter (Becton Dickinson). Cells were resuspended in basic sorting buffer (1× PBS $Ca^{2+}$, and $Mg^{2+}$ free, 5 mM EDTA, 25 mM Hepes, and 2% FBS), and cells expressing red fluorescence signal were sorted to a tube containing DMEM with 10% FBS and then transferred to an appropriate growth flask. The same procedure was used to enrich the double expressing H2B-GFP and mRFP-CLASP2γ cell lines, selecting for cells with double fluorescence signal (GFP and mRFP).

### RNAi

RNAi experiments were performed in $0.2 \times 10^6$ cells cultured in 1.5 ml DMEM with 5% FBS. A solution containing 2 µl of Lipofectamine RNAiMax (Invitrogen) diluted in 250 µl Opti-MEM was mixed with another containing 100 pmol of siRNA (Sigma-Aldrich) diluted in 250 µl Opti-MEM. The solution mix was incubated for 30 min at room temperature and then added dropwise to the cells. The siRNA oligonucleotides used were Scrambled siRNA, 5′-CUUCCUCUCUUUCUCUCCCUUGUGA-3′; CLASP1#A siRNA, 5′-GCCAUUAUGCCAACUAUCU-3′; CLASP2#A siRNA, 5′-GUUCAGAAAGCCCUUGAUG-3′; CLASP1#B siRNA, 5′-GGAUGAUUUACAAGACUGG-3′; and CLASP2#B siRNA, 5′-GACAUACAUGGGUCUUAGA-3′ (Mimori-Kiyosue et al., 2005). After a 6-h incubation at 37°C, the medium was removed and replaced by 2 ml of DMEM with 10% FBS. For Ndc80 depletion, we used a predesigned siRNA oligonucleotide with the sequence 5′-GAAUUGCAGCAGACUAUUA-3′ (Sigma-Aldrich).

### Western blotting

Protein extracts were obtained from cells by addition of lysis buffer (20 mM Hepes/KOH, 1 mM EDTA, 1 mM EGTA, 150 mM NaCl, 0.5% NP-40, 10% glycerol, 2 mM DTT at –20°C, and protease inhibitor 4C+PMSF 0.1 mM at –20°C [1:100], pH 7.9) and frozen with $LN_2$. The suspension was centrifuged for 5 min at 14,000 rpm at 4°C, collecting the supernatant. Total protein levels were quantified using Bradford reagent (Thermo Scientific) and BSA (Thermo Scientific) solutions as standards. 15 µg of total protein per sample were mixed with sample buffer (50 mM Tris-HCl, pH 6.8, 2% SDS, 10% glycerol, 1% β-mercaptoethanol, 12.5 mM EDTA, and 0.02% bromophenol blue) and denatured at 95°C for 5 min. Samples were loaded in a 6.5% acrylamide gel mounted in a Mini-PROTEAN vertical electrophoresis apparatus (Bio-Rad) using an NZYColour Protein Marker II (NZYTech). Blotting was performed with an iBlot Gel Transfer System (Invitrogen). Membranes were blocked with 5% powder milk in PBS Tween 0.1% for 45 min. For primary antibodies, we used hybridoma supernatant of monoclonal antibody 6E3 rat anti-CLASP2 (Maffini et al., 2009) at 1:150, 5F8 rat monoclonal anti-RFP (ChromoTek) at 1:1,000, and mouse monoclonal anti–α-tubulin B-512 (Sigma-Aldrich) at 1:10,000, diluted in 5% powder milk in PBS Tween 0.1%, and incubated overnight (4°C) with agitation. Anti-rat HRP and anti-mouse HRP secondary antibodies (Santa Cruz Biotechnology) were used at 1:2,000 proportions in 5% powder milk in PBS Tween 0.1% and incubated for 1 h. Signal was developed with Clarity Western ECL Blotting Substrate (Bio-Rad) and detected in a Bio-Rad Chemidoc XRS system, with Image Lab software.

## Cold-shock experiments

To induce metaphase arrest, cells were treated with 5 µM MG132 (Merck) for 1 h. Subsequently, cells were placed on ice with ice-cold medium containing 5 µM MG132 and incubated for 7 min before fixation with methanol at –20°C.

## Immunofluorescence

Cells were fixed using PFA at 4% in cytoskeleton buffer (CB; 137 mM NaCl, 5 mM KCl, 1.1 mM Na$_2$HPO$_4$, 0.4 mM KH$_2$PO$_4$, 2 mM EGTA, 2 mM MgCl$_2$, 5 mM Pipes, and 5 mM glucose) or alternatively, in methanol at –20°C, for 10 min (for Nocodazole-arrested cell experiments, Nocodazole was added at final concentration of 3.3 µM to the culture, 2–3 h before fixation; for Mad1 staining experiments in metaphase, cells were treated with 5 µM MG132 for 1 h prior to fixation). Extraction was accomplished using CB–Triton X-100 0.5% for 10 min. Primary antibodies used in this work were mouse anti–α-tubulin clone B-512 at 1:1,500 (Sigma-Aldrich), mouse monoclonal anti-EB1 at 1:500 (BD Biosciences, clone 5/EB1), human ACAs at 1:1,000 (Fitzgerald Industries International, 90C-CS1058), 5F8 rat monoclonal anti-RFP at 1:1,000 (ChromoTek), and mouse anti-Mad1 clone BB3-8 at 1:500 (Millipore) diluted in PBS–Triton X-100 0.1% with 10% FBS. Secondary antibodies Alexa Fluor anti-mouse 488, anti-rat 568, and anti-human 647 (Invitrogen) were diluted to 1:1,000 in 0.1% PBS–Triton X-100 with 10% FBS. DNA staining was achieved by addition of 1 µg/ml DAPI (Sigma-Aldrich). Coverslips were mounted using mounting medium (20 mM Tris, pH 8, 0.5 mM N-propyl gallate, and 90% glycerol). Images were acquired with an Axio imager Z1 (Carl Zeiss) equipped with an ORCA-R2 precooled charge-coupled device (CCD; Hamamatsu), using an immersion oil 63× 1.46 NA plan-apochromatic objective lens, controlled by Zen software. 3D deconvolution was performed using AutoQuant X Image Deconvolution software (Media Cybernetics). Histogram adjustment and image quantifications were performed with ImageJ software.

## Live-cell imaging

The mRFP-CLASP2γ U2OS cells expressing H2B-GFP were cultured in glass coverslips using DMEM without phenol red and supplemented with 25 mM Hepes (Gibco) and 10% FBS. SiR-tubulin (Spirochrome) was added to a final concentration of 20 nM 30–60 min before observation. Time-lapse imaging was performed in a heated chamber (37°C) using a 60× oil-immersion 1.40 NA plan-apochromatic objective mounted on an inverted microscope (Eclipse TE2000U; Nikon) equipped with a CSU-X1 spinning-disk confocal head (Yokogawa Corporation of America) controlled by NIS-Elements software and with three laser lines (488 nm, 561 nm, and 647 nm). Images were detected with an iXonEM+ EM-CCD camera (Andor Technology). 11 z-planes separated by 1 µm were collected every 2 min. Histogram adjustment and image quantifications were performed using ImageJ software and NIS Viewer (Nikon).

## Photoactivation experiments

MT turnover rates were measured in the mRFP-CLASP2γ U2OS cells stably expressing the PA–GFP–α-tubulin. Cells were grown in glass coverslips with DMEM without phenol red and with 25 mM of Hepes (Gibco), supplemented with 10% FBS. Time-lapse imaging was performed in a heated chamber (37°C) using a 100× 1.4 NA plan-apochromatic differential interference contrast (DIC) objective mounted on an inverted microscope (Nikon TE2000U) equipped with a CSU-X1 spinning-disk confocal head (Yokogawa Corporation of America) and with two laser lines (488 nm and 561 nm). Images were detected with an iXonEM+ EM-CCD camera (Andor Technology). Photoactivation was performed with a Mosaic digital mirror device–based patterning system (Andor) equipped with a 405-nm diode laser. Cells in metaphase were chosen using DIC acquisition. A thin stripe of spindle MTs was locally photoactivated in one half-spindle by pulsed near-UV irradiation (405-nm laser; 500-µs exposure time) and fluorescence images (seven 1-µm separated z-planes centered at the middle of the mitotic spindle) were captured every 15 s for 4.5 min with a 100× oil-immersion 1.4 NA plan-apochromatic objective. To quantify fluorescence dissipation after photoactivation, spindle poles were aligned horizontally, and whole-spindle, sum-projected kymographs were generated (sum projections using ImageJ and kymographs generated as previously described [Pereira and Maiato, 2010]). Fluorescence intensities were quantified for each time point (custom-written routine in Matlab, "LAPSO" software) and normalized to the first time point after photoactivation for each cell following background subtraction and correction for photobleaching. Correction for photobleaching was performed by normalizing to the values of fluorescence loss obtained from whole-cell (including the cytoplasm), sum-projected images for each individual cell. This method allows for the precise measurement of photobleaching for each cell at the individual level and avoids any potentially unspecific Taxol-associated phenotypes. Under these conditions, the photoactivated region dissipates, yet the photoactivated molecules are retained within the cellular boundaries defined by the cytoplasm. To calculate MT turnover, the sum intensity at each time point was fit to a double exponential curve $A1*exp(-k_1*t) + A2*exp(-k_2*t)$ using Matlab (Mathworks), in which $t$ is time, $A1$ represents the less stable (non-KT–MT) population, and $A2$ the more stable (KT–MT) population with decay rates of $k_1$ and $k_2$, respectively. When running the routine to fit the data points to the model curve, the rate constants are obtained as well as the percentage of MTs for the fast (typically interpreted as the fraction corresponding to non-KT–MTs) and the slow (typically interpreted as the fraction corresponding to KT–MTs) processes. The half-life for each process was calculated as $ln2/k$ for each population of MTs. All experiments were performed in the presence of MG132 (5 µM), which was added 30 min before imaging to ensure that cells were in metaphase and prevent mitotic exit. Spindle poleward flux rates were determined in the same cells used for quantification of spindle MT half-life by determining the slope of the fluorescence signal over time (4.5 min), using a Matlab custom-written routine in LAPSO software (Afonso et al., 2019) and using the slope of the movement of the metaphase plate as reference.

## Statistical analysis

Statistical analysis was performed with Graphpad Prism, version 8. The statistical significance of KT colocalization, spindle

length, segregation errors, NEB to anaphase, and NEB to metaphase durations among all experimental conditions was determined using the parametric unpaired Student's $t$ test. The statistical significance of both KT–MT and non-KT–MT half-lives and spindle poleward flux was determined using the nonparametric Mann-Whitney U-Test. The statistical significance of the cold shock and Mad1-positive cells was determined using parametric unpaired Welch's $t$ test.

### Online supplemental material

Fig. S1 shows the influence of the purification tags and the amount of injected protein in the elution profile of the recombinant CLASP2 C-terminal, determined by SEC. Fig. S2 shows Western blots confirming the expression levels of the mRFP-CLASP2γ constructs and the efficiency of endogenous CLASP depletion by RNAi in all cell lines. Fig. S3 shows localization of all mRFP-CLASP2γ constructs in metaphase cells. Fig. S4 shows the colocalization of the mRFP-CLASP2γ constructs with EB1 at the MT plus-ends in interphase cells. Fig. S5 shows representative decay curves, spindle MT half-life values, and the percentage ratio of the two spindle MT populations obtained in the photoactivation experiments. Video 1 shows representative mitotic progression of human parental U2OS cell lines stably expressing PA–GFP–α-tubulin and H2B-GFP, without and with CLASPs RNAi. Videos 2, 3, 4, 5, 6, 7, 8, 9, and 10 show representative mitotic progression and chromosome segregation errors in the different mRFP-CLASP2γ–expressing cell lines upon endogenous CLASP depletion by RNAi.

### Acknowledgments

We thank Michael Lampson for kindly providing the Spc25 plasmid, and Bernardo Orr, Jorge Ferreira, and António Pereira for the critical reading of this manuscript.

H. Girão was supported by a studentship from Fundação para a Ciência e a Tecnologia (SFRH/BD/141066/2018). This work was supported by a Grant-in-Aid for Scientific Research 22570190 (to I. Hayashi) and by a grant agreement from the European Research Council (681443) under the European Union's Horizon 2020 research and innovation program (to H. Maiato). The authors acknowledge the support of the B2Tech i3S Scientific Platform.

The authors declare no competing financial interests.

Author contributions: H. Girão performed and analyzed all the experiments, except Figs. 1, 6, and S1, under supervision by N. Okada and H. Maiato. N. Okada performed and analyzed the experiments reported in Fig. 6, and generated tools. T.A. Rodrigues, A.O. Silva, A.C. Figueiredo, and Z. Garcia performed the biochemical characterization of CLASP2, under supervision of S. Macedo-Ribeiro, J.E. Azevedo, and H. Maiato. I. Hayashi provided tools. T. Moutinho-Santos generated tools and preliminary data. H. Maiato conceived and coordinated the project, designed experiments, and analyzed the data. H. Girão and H. Maiato wrote the paper, with contributions from all the authors.

Submitted: 10 May 2019

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
