## [Reviewer comments · The Journal of Cell Biology]

CLASP2 binding to curved microtubule tips promotes flux and stabilizes kinetochore attachments

Hugo Girão, Naoyuki Okada, Tony Rodrigues, Alexandra Silva, Ana Figueiredo, Zaira Garcia, Tatiana Moutinho-Santos, Ikuko Hayashi, Jorge Azevedo, Sandra Macedo Ribeiro, and Helder Maiato

Corresponding Author(s): Helder Maiato, Institute for Molecular and Cell Biology University of Porto

Review Timeline:

Submission Date:	2019-05-10
Editorial Decision:	2019-06-14
Revision Received:	2019-10-17
Editorial Decision:	2019-10-29
Revision Received:	2019-10-31

Monitoring Editor: Rebecca Heald

Scientific Editor: Melina Casadio

Transaction Report:

DOI: <https://doi.org/10.1083/jcb.201905080>

June 14, 2019

Re: JCB manuscript #201905080

Prof. Helder Maiato
Institute for Molecular and Cell Biology University of Porto
Rua Alfredo Allen, 208
Porto 4200-135
Portugal

Dear Helder,

Thank you for submitting your manuscript entitled "CLASP2 lattice-binding near microtubule plus ends stabilizes kinetochore attachments". The manuscript was assessed by expert reviewers, whose comments are appended to this letter. We invite you to submit a revision if you can address the reviewers' key concerns, as outlined here.

You will see that all three reviewers find your study to be well executed and potentially interesting. However, a number of concerns need to be addressed. Reviewers #1 and #2 both request a more nuanced analysis of CLASP complex size fractionation. Reviewer #2 suggests an experiment to drive CLASP kinetochore localization in the absence of dimerization, which I think would be a great addition to your study if it is not too onerous. Both Reviewers #2 and #3 request a more complete description of chromosome segregation defects. Reviewer #3 would also like to see more analysis of microtubule turnover in both metaphase and prometaphase spindles. While I agree that it would be nice to include these data if you already have them, I am not convinced that performing this analysis would add substantially to the conclusions of your study and therefore we do not insist that you address this point experimentally. However, it would be important to know whether kinetochore fibers are forming properly. In addition, I hope you will be able to address the minor points raised by the reviewers by adding additional quantification and/or changes to the text.

GENERAL GUIDELINES:

Text limits: Character count for a Report is < 20,000, not including spaces. Count includes title page, abstract, introduction, results, discussion, acknowledgments, and figure legends. Count does not include materials and methods, references, tables, or supplemental legends.

Figures: Reports may have up to 5 main text figures. To avoid delays in production, figures must be prepared according to the policies outlined in our Instructions to Authors, under Data Presentation, <http://jcb.rupress.org/site/misc/ifora.xhtml>. All figures in accepted manuscripts will be screened prior to publication.

***IMPORTANT: It is JCB policy that if requested, original data images must be made available. Failure to provide original images upon request will result in unavoidable delays in publication. Please ensure that you have access to all original microscopy and blot data images before

submitting your revision.***

Supplemental information: There are strict limits on the allowable amount of supplemental data. Reports may have up to 3 supplemental figures. Up to 10 supplemental videos or flash animations are allowed. A summary of all supplemental material should appear at the end of the Materials and methods section.

Our typical timeframe for revisions is three months; if submitted within this timeframe, novelty will not be reassessed at the final decision. Please note that papers are generally considered through only one revision cycle, so any revised manuscript will likely be either accepted or rejected.

Thank you for this interesting contribution to the Journal of Cell Biology. You can contact us at the journal office with any questions, cellbio@rockefeller.edu or call (212) 327-8588.

Sincerely,

Rebecca Heald, Ph.D.
Monitoring Editor, Journal of Cell Biology

Melina Casadio, Ph.D.
Senior Scientific Editor, Journal of Cell Biology

Reviewer #1 (Comments to the Authors (Required)):

The control of microtubule dynamics at the kinetochore interface is critical for proper chromosome alignment and segregation. The CLASP proteins are important players in modulating this dynamic interface, but the detailed molecular mechanism of their interactions have been controversial. This is attributed to the finding that there are multiple CLASP isoforms in humans, and in addition, there are multiple functional domains that have been reported to have slightly different functions in different organisms. In the current study, the authors use a knockdown/rescue strategy in which they express CLASP2 Δ in which different functional domains have been mutated. This is complemented by a biochemical analysis of the endogenous CLASP2 in cultured cell extracts. They find that CLASP2 is dimeric, which is different than previous reports, which stated that it was a monomer. The dimerization is mediated by the C-terminal domain, which also contains the kinetochore targeting sequence. Artificial dimerization is not sufficient to rescue kinetochore localization, spindle length, mitotic progression, nor the fidelity of chromosome segregation. In addition, mutation of the EB1 binding domains or the TOG2/3 domains also prevents rescue of any of the mitotic phenotypes.

Overall, this is a beautifully executed study with rigorous quantification of all phenotypes that is clearly presented to the reader. I would like to thank the authors for not making me work hard to figure out what you did and what you counted!!! My life as a reviewer would be much simpler if

others would present their experiments so clearly. One question arises as to the significance of the advance. On one hand, it would be more exciting if they were able to generate separation of function mutants in which each mutant rescued different aspects of CLASP function, but the results are what they are, which states that all of the functional domains contribute to the complex activity of CLASP in the spindle. This is important to know. In addition, because this is a very carefully done study that quantitatively assesses multiple phenotypes with good re-expression of the mutant proteins, it should provide the benchmark for other studies of this type. For these reasons, I support publication of this work as a report in JCB after the authors address the comments below.

1. I disagree with some of the authors' conclusions regarding the size analysis of CLASP. In the sucrose gradient, all of the fractions are adjacent so there are not two distinguishable pools of protein. For gel filtration, there is some protein in the void, a peak in fraction 9, a peak in fraction 13, again in 15-17 and then in 19- so it is apparent throughout the column. The authors have selected only some of these pools in the quantification of overall size, which is not appropriate. One issue may be that the authors had to precipitate the protein before running on the gel, which may create some irregularities. As a suggestion, these assays are more quantitative if you do them by ELISA rather than by gel and blot. The ELISA can be read on a plate reader, and then the signal can be plotted across the fraction. In either case, the authors must present data that more clearly illustrates just two major pools of protein to support their conclusion.
2. In Figure S1, the Delta-CT and the GCN4 constructs are expressed to a higher level than others. They still don't rescue, so this does not change the conclusions of the study, but I disagree that the levels of expression are all comparable. Please change the wording to reflect the actual differences.
3. It would be helpful if the authors more clearly stated the distinctions from other studies throughout the results and discussion and then summarized it at the end. It wasn't until the very last two paragraphs that I fully understood what was unique about the current work. This is a short paper with results and discussion combined, so it would be helpful if the authors took advantage of that format to highlight their work.

Reviewer #2 (Comments to the Authors (Required)):

In this manuscript, Girão et al. describe how three functional domains of the human CLASP2, a conserved microtubule associated protein, coordinate to regulate mitosis. Through multiple biochemical approaches, they demonstrate that physiologically relevant CLASP2 exists as both monomer and dimers similar to the yeast ortholog, yet unlike the yeast protein CLASP2 does not appear to bind free tubulin. This analysis clears up confusion about which protein activities have been evolutionarily conserved as many CLASPs studies have focused on the fungal and xenopus experimental systems. Additionally, they use high resolution live- and fixed-cell microscopy to characterize three mutants (alone or in combination) that disrupt CLASP2 localization to (i) microtubule plus ends, (ii) microtubule tips, and (iii) kinetochores. From these analyses, they conclude that all three of these activities cooperatively regulate kinetochore-microtubule attachments to ensure accurate chromosome segregation.

All experiments in this study are well-executed. The data are well documented, often analyzed through numerous techniques, and appropriate controls were provided in the supplementary information. Their robust analysis using the human protein clears up confusion/assumptions about this family of proteins, particularly the ability to bind free tubulin vs. microtubules. In addition, their work advances the field by showing that kinetochore localization is required for the regulation of

kinetochore-microtubule dynamics. The one thing the study would benefit from is integrating the biochemical and cell biology approaches in the discussion. While a key finding was that artificial homodimerization of CLASP2 failed to rescue any of the mitotic phenotypes, the authors did not address why dimerization is required and why this activity would be conserved in CLASP2. They do not propose how they envision the three distinct activities integrating at kinetochores and why defects in the activities lead to chromosome segregation errors. If the authors are able to put the work into a broader context with a revised discussion, the paper should be of general interest to cell biologists.

Major Comments

1. Studying the endogenous CLASP2 in cell extracts provided excellent physiologically relevant results. I'm convinced cells form some sort of multimeric complex containing CLASP2. However, a number of curiosities (primarily that artificial dimerization fails to rescue all phenotypes) suggest a bit more study should be done to better define the complexes they observed.
 - a. In Fig 1B, the SEC profile labels fractions 12-17 as "Pool 1" yet to my eye it seems possible that fraction 12-14 represent one population and 15-17 another.
 - b. Monomeric CLASP2 was calculated to be 153 kDa while the dimeric species was calculated to be 273 kDa. The dimer should be observed closer to 300 kDa especially considering its large frictional coefficient is likely to cause an overestimation of its mass.
 - c. Recombinant CLASP2 C-terminus purified as a single dimeric peak yet in cells CLASP2 exists as a monomer as well suggesting some sort of regulation between these two states.

While not all of these need to be addressed, I do believe staining the SEC fractions for EB1/2/3 and CLASP1 would help understand the possible composition of higher molecular weight species.

Additionally, targeting the monomeric ΔC construct to the kinetochore through a protein fusion or the FKBP system would also bolster the findings here and would achieve kinetochore localization independent of dimerization. If this mutant rescues the phenotypes it would indicate a unique structure of the CLASP2 dimerization is required for kinetochore binding, but not existing as a dimer per se; if it fails that would suggest CLASP2 uses this domain to interact with a different binding partner. Either result should better inform the role of the dimerization domain.

2. If the authors are going to claim CLASP2 has a role in error correction, they should better describe the chromosome segregation errors. Are these merotelic attachments resulting in lagging chromosomes? How do they envision weaker kinetochore-microtubule attachments resulting in more merotelic attachments? Is it similar to Kif18a loss where single chromosomes migrate slower in anaphase and then are excluded from the reassembled nuclear envelope?

Minor Comments

1. It was not clear why CLASP2 γ was used when CLASP2 α is the predominant species in the U2OS cells. Additionally, it would be helpful to mention the differences between the isoforms more specifically and sooner. The only description was the absence of TOG1 and that was mentioned ~150 lines after the first mutant was introduced.
2. Is the fluorescence decay assay sensitive enough to measure changes to dynamics of non-kinetochore microtubules? The acquisition time for those experiments was every 15 seconds but the half-life of this population is 10 seconds. Clearly this population could be detected, but it may not pick up slight changes due to CLASP loss without shorter acquisition times.
3. In Sup Fig 1 and 2 both of the constructs with the 2ea-3eeaa mutation seem to express at

slightly lower levels - could this explain the loss of kinetochore signal with these mutants?

Reviewer #3 (Comments to the Authors (Required)):

In this manuscript the authors report results of experiments designed to understand the contribution of CLASP2 to mitosis in mammalian cells. CLASP proteins are known to contribute to mitotic fidelity, spindle bipolarity and poleward microtubule flux. The main findings presented here are that kinetochore binding, TOG domains and SxIP motifs all contribute to CLASP function in mitosis and that CLASP2 can form dimers in vitro.

The authors first demonstrate that the C-terminal region of CLASP2 mediates dimerization. In solution, the isolated C-terminal fragment shows behavior (size exclusion chromatography and density gradient centrifugation) consistent with dimerization. The full-length protein analyzed on native electrophoretic gels shows species consistent with monomers, dimers and even larger species; sucrose gradients and chromatography also show evidence of dimers and monomeric species. The behavior of the C-terminal part differs from the full-length protein as only one species (dimer) is found. This suggests that other regions of the full-length protein (not just protein concentration (page 8) could regulate dimerization. Similar results have been reported for Human CLASP1.

Most of the manuscript describes fixed and live cell analysis of cells which are depleted of endogenous CLASP2 using siRNA targeting the endogenous protein and that express various CLASP2 mutants -- including a mutated SxIP motif, two TOG domains mutated, a construct with both SxIP and TOG mutated, as well as CLASP2 lacking the C-terminal domain and a construct that has an artificial dimerization motif in place of the C-terminal region.

The localization to the spindle and microtubule tips is very weak, suggesting that CLASP2 mediates its effects primarily at the kinetochore. The authors describe the different domains of CLASP2 as independently contributing. In the context of the spindle, the kinetochore pool seems to be most important, so this might be clarified and discussed.

Cells with the various mutations show defects in chromosome segregation but the nature of the defects is not clear. Movies show chromosomes that lag, but no analysis of the types of segregation defects was provided.

Microtubule turnover is measured in cells expressing PA-GFP tubulin. The results show that the half-time for turnover is faster in cells expressing the mutant CLASP2 proteins and depleted of endogenous CLASPs, with a more significant change when the two C-terminal mutant versions of CLASP2 are expressed. The authors say that the cells were treated with MG132 for 30min, so they should be in metaphase. However, prior work has shown that kinetochore microtubule turnover is faster in prometaphase cells and becomes reduced after the last chromosome has aligned at the metaphase plate (Kebeche and Compton; please add the citation). Could the change in turnover result because the cells expressing mutant CLASP take longer to progress to metaphase, and therefore are not in full metaphase when the measurements are performed? The authors should measure turnover in PM cells as well as metaphase cells and determine if the stage of mitosis impacts the measurements.

A related concern is that the PA-tubulin marks in some of the cells appear to lack bright

Kinetochores fiber microtubule bundles, suggesting that k-fibers do not form when CLASP is mutant/absent. The cells should be stained for a check-point protein to determine if they generate stable attachments, and satisfy the checkpoint. The photoactivation data are fit to a double exponential which reports on the k-fiber and non-K microtubule populations; however if kinetochores fibers do not form or are altered how are the data analyzed-- are both fast and slow populations present? Representative graphs of fluorescence dissipation should be shown, and the authors should provide a Table with the values for the percent of the Photoactivated microtubules that are in the fast and slow populations, and the half-time values. The rates of flux should also be provided, as it would be of interest to know if that parameter is also altered.

Other:

In figure 2 the intensity of kinetochores CLASP2 appears lower in the cells expressing IP12 mutant CLASP, as compared to the CLASP with mutations in TOG domains (2ea-3eeaa) or CLASP2 with mutations in both SxIP and TOG domains. Is the image representative?

There is no quantification of the western blots. The cell line expressing IP12-3eeaa mutant CLASP2 has (apparently) a much reduced level of protein. This is worrisome because the lower level of protein could contribute to the phenotype in these cells.

Supplemental figure 3. Imaging CLASP at microtubule plus-ends is weak. The cells expressing wild-type CLASP do show co-localization, but in all of the other panels, the CLASP images look very punctate without clear "dashes" (perhaps with the exception of the "dash" that is highlighted in the box insets. Even in the nuclear area, where the background is low, the colocalization is not obvious in the images supplied. This makes the evaluation of the mutants very challenging. Can the authors either provide better images or discuss the evidence showing that CLASP2 tip-tracks in mitosis?

Similarly, in supplemental figure 3B, the spindle staining is very weak - hardly above background, making the data hard to evaluate. The 2ea-3eeaa expressing cells have weak spindle fluorescence, but also fewer microtubules in the tubulin channel. It is very hard to say if the IP12-3eeaa mutant has any specific staining at all.

Page 14: the authors discuss figure 5, which shows the PA of tubulin and measurement of turnover half-time. They state that the change in dynamic turnover was "exacerbated by the combined mutation of the TOG3 and SxIP domains" but in Fig 5B, the turnover of Kinetochores microtubules does not appear to be different for the double mutant, as compared to either of the constructs with a single mutation.

Movie 1 siRNA CLASP2 is the chromosome lagging or did it never attach?

Movie 6: is the spindle rocking/oscillating typical for cells expressing this mutant?

Consider changing incapacity to inability.

Golden standard should be gold standard.

Point-by-point response (our comments in blue)

Reviewer #1 (Comments to the Authors (Required)):

The control of microtubule dynamics at the kinetochore interface is critical for proper chromosome alignment and segregation. The CLASP proteins are important players in modulating this dynamic interface, but the detailed molecular mechanism of their interactions have been controversial. This is attributed to the finding that there are multiple CLASP isoforms in humans, and in addition, there are multiple functional domains that have been reported to have slightly different functions in different organisms. In the current study, the authors use a knockdown/rescue strategy in which they express CLASP2 γ in which different functional domains have been mutated. This is complemented by a biochemical analysis of the endogenous CLASP2 in cultured cell extracts. They find that CLASP2 is dimeric, which is different than previous reports, which stated that it was a monomer. The dimerization is mediated by the C-terminal domain, which also contains the kinetochore targeting sequence. Artificial dimerization is not sufficient to rescue kinetochore localization, spindle length, mitotic progression, nor the fidelity of chromosome segregation. In addition, mutation of the EB1 binding domains or the TOG2/3 domains also prevents rescue of any of the mitotic phenotypes.

Overall, this is a beautifully executed study with rigorous quantification of all phenotypes that is clearly presented to the reader. I would like to thank the authors for not making me work hard to figure out what you did and what you counted!!! My life as a reviewer would be much simpler if others would present their experiments so clearly. One question arises as to the significance of the advance. On one hand, it would be more exciting if they were able to generate separation of function mutants in which each mutant rescued different aspects of CLASP function, but the results are what they are, which states that all of the functional domains contribute to the complex activity of CLASP in the spindle. This is important to know. In addition, because this is a very carefully done study that quantitatively assesses multiple phenotypes with good re-expression of the mutant proteins, it should provide the benchmark for other studies of this type. For these reasons, **I support publication of this work as a report in JCB** after the authors address the comments below.

R: We thank the reviewer for the encouraging words and for supporting publication of our work. We hope to have addressed all pending issues in this revised version.

1. I disagree with some of the authors' conclusions regarding the **size analysis of CLASP**. In the sucrose gradient, all of the fractions are adjacent so **there are not two distinguishable pools of protein**. For gel filtration, there is some protein in the void, a peak in fraction 9, a peak in fraction 13, again in 15-17 and then in 19- so it is apparent throughout the column. **The authors have selected only some of these pools in the quantification of overall size, which is not appropriate**. One issue may be that the authors had to precipitate the protein before running on the gel, which may create some irregularities. As a suggestion, these assays are more quantitative if you do them by ELISA rather than by gel and blot. The ELISA can be read on a plate reader, and then the signal can be plotted across the fraction. In either case, **the authors must present data that more clearly illustrates just two major pools of protein to support their conclusion**.

R: We thank the reviewer for drawing our attention to important details on the biochemical analysis of CLASP2. After re-inspection of our original data, which, admittedly, were collected in our lab over many years and involving different experimenters, we agree with the main points raised by the reviewer, particularly regarding the distinction of two major protein pools in the sucrose gradient. For this reason, we have decided to re-do and extend all the biochemical analysis from scratch (including recombinant full-length CLASP2, not included in the original submission) under strong in-house supervision by highly experienced biochemists and structural biologists. Our findings now indicate that although CLASP2 has a weak potential to self-associate through its C-terminal domain, our new data offers no doubts that CLASP2 exists predominantly as an elongated monomer in solution. This is further supported by new functional experiment that showed that driving monomeric CLASP2 to kinetochores was able to fully rescue normal kinetochore-microtubule dynamics and partially rescued all mitotic phenotypes resulting from endogenous CLASPs depletion.

2. In Figure S1, the Delta-CT and the GCN4 constructs are expressed to a higher level than others. They still don't rescue, so this does not change the conclusions of the study, but I disagree that the levels of expression are all comparable. Please **change the wording to reflect the actual differences**.

R: We have rewritten this section to address the point raised by the reviewer.

3. It would be helpful if the authors **more clearly stated the distinctions from other studies throughout the results and discussion and then summarized it at the end**. It wasn't until the very last two paragraphs that I fully understood what was unique about the current work. This is a short paper with results and discussion combined, so it would be helpful if the authors took advantage of that format to highlight their work.

R: In order to accommodate the new data and clarify the distinctions from other studies (as pointed out by the reviewer) we have now changed the manuscript format to full article and have extensively re-written the results and discussion as separate sections.

Reviewer #2 (Comments to the Authors (Required)):

In this manuscript, Girão et al. describe how three functional domains of the human CLASP2, a conserved microtubule associated protein, coordinate to regulate mitosis. Through multiple biochemical approaches, they demonstrate that **physiologically relevant CLASP2 exists as both monomer and dimers similar to the yeast ortholog, yet unlike the yeast protein CLASP2 does not appear to bind free tubulin**. This analysis clears up confusion about which protein activities have been evolutionarily conserved as many CLASPs studies have focused on the fungal and xenopus experimental systems. Additionally, they use high resolution live- and fixed-cell microscopy to characterize three mutants (alone or in combination) that disrupt CLASP2 localization to (i) microtubule plus ends, (ii) microtubule tips, and (iii) kinetochores. From these analyses, they conclude that all three of these activities cooperatively regulate kinetochore-microtubule attachments to ensure accurate chromosome segregation.

All experiments in this study are well-executed. The data are well documented, often analyzed through numerous techniques, and appropriate controls were provided in the supplementary information. Their

robust analysis using the human protein clears up confusion/assumptions about this family of proteins, particularly **the ability to bind free tubulin vs. microtubules**. In addition, **their work advances the field by showing that kinetochore localization is required for the regulation of kinetochore-microtubule dynamics**. The one thing the study would benefit from is **integrating the biochemical and cell biology approaches in the discussion**. While a key finding was that artificial homodimerization of CLASP2 failed to rescue any of the mitotic phenotypes, **the authors did not address why dimerization is required and why this activity would be conserved in CLASP2**. They do not **propose how they envision the three distinct activities integrating at kinetochores and why defects in the activities lead to chromosome segregation errors**. If the authors are able to **put the work into a broader context with a revised discussion**, the paper should be of general interest to cell biologists.

R: We thank the enthusiasm and insightful comments by the reviewer and his/her guidance concerning how to improve the communication, interpretation and discussion of the data. In this revised manuscript, we have also re-done all the biochemistry from scratch (please see our response to reviewer #1) and, together with new functional data, our results favor the formation of human CLASP2 monomers as the predominantly active species in cells. Our data also clearly indicates that binding to soluble tubulin is fully dispensable for mitosis in human cells. We have also now included a model illustrating how we envision the integration of the distinct CLASP2 activities at the kinetochore-microtubule interface. Finally, we have extensively revised our Discussion, now presented in a separate section due to manuscript reformatting into full Article.

Major Comments

1. Studying the endogenous CLASP2 in cell extracts provided excellent physiologically relevant results. I'm convinced cells form some sort of multimeric complex containing CLASP2. However, a number of curiosities (primarily that artificial dimerization fails to rescue all phenotypes) suggest a bit more study should be done to better define the complexes they observed.

a. In Fig 1B, the SEC profile labels fractions 12-17 as "Pool 1" yet to my eye it seems possible that fraction 12-14 represent one population and 15-17 another.

b. Monomeric CLASP2 was calculated to be 153 kDa while the dimeric species was calculated to be 273 kDa. The dimer should be observed closer to 300 kDa especially considering its large frictional coefficient is likely to cause an overestimation of its mass.

c. Recombinant CLASP2 C-terminus purified as a single dimeric peak yet in cells CLASP2 exists as a monomer as well suggesting some sort of regulation between these two states.

While not all of these need to be addressed, I do believe staining the SEC fractions for EB1/2/3 and CLASP1 would help understand the possible composition of higher molecular weight species.

R: We agree with all the points and have addressed them as a whole by completely re-investigating CLASP2 biochemistry, both for recombinant C-terminal and now for full-length proteins, as well as for endogenous CLASP2. In addition to SEC and sedimentation gradient centrifugation, we now include new dynamic light scattering and isothermal titration calorimetry data. The results are consistent with CLASP2 being predominantly a monomer in solution that does not bind soluble tubulin in cells, but it has the capacity to self-associate. Importantly, by including a new CLASP2 mutant that is driven to kinetochore by fusion with Spc25 (see next point) and is able to fully rescue normal kinetochore-microtubule dynamics in the absence of endogenous CLASPs, clearly indicates that CLASP2 fulfills its kinetochore function independently of self-association, while linking the biochemical and functional data.

Additionally, targeting the monomeric ΔC construct to the kinetochore through a protein fusion or the FKBP system would also bolster the findings here and would achieve kinetochore localization independent of dimerization. If this mutant rescues the phenotypes it would indicate a unique structure of the CLASP2 dimerization is required for kinetochore binding, but not existing as a dimer per se; if it fails that would suggest CLASP2 uses this domain to interact with a different binding partner. Either result should better inform the role of the dimerization domain.

R: This was a great suggestion of a win/win experiment that turned out to give us a decisive answer, while helping to re-interpret the CLASP2 biochemistry now re-investigated and re-interpreted in the revised manuscript. Among the different possibilities, we chose to fuse CLASP2 ΔC with Spc25 (monomeric protein, part of the Ndc80 complex), for which we had all necessary tools available in the lab, and has been used successfully in the past to ectopically/constitutively target CENP-E (a CLASP-recruiting factor – Maffini et

al., Curr Biol, 2009) to kinetochores in human cells (Zhang et al., Nature Chem Biol, 2017). Establishment of a stable cell line expressing this construct revealed an ~2 fold increase in CLASP2 levels at unattached kinetochores compared to WT CLASP2, consistent with the constitutive localization of Spc25 at kinetochores. Importantly, this construct completely rescue normal kinetochore microtubule half-life and flux after endogenous CLASP1/2 depletion, while partially rescuing all the mitosis parameters evaluated (spindle length, mitotic duration, chromosome segregation errors). We conclude that CLASP2 dimerization is largely dispensable for kinetochore function during mitosis.

2. If the authors are going to claim CLASP2 has a role in error correction, they should better describe the chromosome segregation errors. Are these merotelic attachments resulting in lagging chromosomes? How do they envision weaker kinetochore-microtubule attachments resulting in more merotelic attachments? Is it similar to Kif18a loss where single chromosomes migrate slower in anaphase and then are excluded from the reassembled nuclear envelope?

R: This is a valid point. We would like to highlight that the errors might actually result from the role of CLASP2 in prometaphase, which we are not directly assessing in this work. Nevertheless, we better classified the type of errors observed in our live cell recordings (e.g. lagging chromosomes vs. chromosome bridges) and determined their fate (i.e. whether they resolve and whether or not they reintegrate the main nuclei or instead form micronuclei). More dedicated higher resolution studies in fixed material will be required to specifically address the origin of those errors, but we feel this is beyond the scope of the present study. This is now extensively discussed in the revised manuscript.

Minor Comments

1. It was not clear why CLASP2 γ was used when CLASP2 α is the predominant species in the U2OS cells. Additionally, it would be helpful to **mention the differences between the isoforms more specifically and sooner**. The only description was the absence of TOG1 and that was mentioned ~150 lines after the first mutant was introduced.

R: Given the indistinguishable localization of CLASP2 α and CLASP2 γ isoforms during mitosis, we suspected that the latter might be fully functional. Indeed, in our preliminary observations early on when we started this project we noticed that CLASP2 γ was able to fully rescue CLASP1/2 depletion in U2OS cells and, as so, we decided to work with this isoform. The advantage of using this isoform is the ability to test the requirement of the TOG1 domain, thought to bind to soluble tubulin and whose role remained controversial. As shown clearly in the different experiments, TOG1 is completely dispensable for CLASP2 function in mitosis. We also now explain the different isoform upfront in the manuscript and explain our rationale for using the CLASP2 γ isoform.

2. Is the fluorescence decay assay sensitive enough to measure changes to dynamics of non-kinetochore microtubules? The acquisition time for those experiments was every 15 seconds but the half-life of this population is 10 seconds. Clearly this population could be detected, but it may not pick up slight changes due to CLASP loss without shorter acquisition times.

R: The reviewer has a good point. We are indeed limited to the sensitivity of the method and although we could not find significant differences in non-kMT dynamics in all original mutant conditions, we cannot exclude that minor changes do exist. In fact, one of the new mutants introduced in this revised version did show statistically significant differences, although probably without biological relevance. We have rewritten this section to better account for these possibilities.

3. In Sup Fig 1 and 2 both of the constructs with the 2ea-3eaa mutation seem to express at slightly lower levels - could this explain the loss of kinetochore signal with these mutants?

R: We have now calculated the relative expression levels for all the CLASP2 constructs and, with the noticeable exception of the ΔC construct, which was overexpressed by 64%, we found a variability of approximately 25% (+ or -) relative to controls. This indeed falls within the observed reduction at the kinetochore for the specific mutants that reviewer alluded to and we now discuss this possibility in the main text.

Reviewer #3 (Comments to the Authors (Required)):

In this manuscript the authors report results of experiments designed to understand the contribution of CLASP2 to mitosis in mammalian cells. CLASP proteins are known to contribute to mitotic fidelity, spindle bipolarity and poleward microtubule flux. The main findings presented here are that kinetochore binding, TOG domains and SxIP motifs all contribute to CLASP function in mitosis and that CLASP2 can form dimers in vitro.

The authors first demonstrate that the C-terminal region of CLASP2 mediates dimerization. In solution, the isolated C-terminal fragment shows behavior (size exclusion chromatography and density gradient centrifugation) consistent with dimerization. The full-length protein analyzed on native electrophoretic gels shows species consistent with monomers, dimers and even larger species; sucrose gradients and chromatography also show evidence of dimers and monomeric species. **The behavior of the C-terminal part differs from the full-length protein as only one species (dimer) is found. This suggests that other regions of the full-length protein (not just protein concentration (page 8) could regulate dimerization. Similar results have been reported for Human CLASP1.**

R: Unfortunately, we were wrong regarding the dimerization of CLASP2, but fortunately, after careful re-investigation instigated by the reviewers, we were able to prevent the perpetuation of this mistake. The likely elongated form of CLASP2 accounted for our misinterpretation of the original data and we can now very securely state that CLASP2 exists predominantly as a monomer in solution, despite having the capacity to self-associate through its C-terminal. A complete re-investigation of CLASP2 biochemistry is now included in the revised manuscript, including a new comparison between recombinant CLASP2 C-terminal with recombinant CLASP2 full-length. Peer-review did its job and we are thankful for that.

Most of the manuscript describes fixed and live cell analysis of cells which are depleted of endogenous CLASP2 using siRNA targeting the endogenous protein and that express various CLASP2 mutants -- including a mutated SxIP motif, two TOG domains mutated, a construct with both SxIP and TOG mutated, as well as CLASP2 lacking the C-terminal domain and a construct that has an artificial dimerization motif in place of

the C-terminal region. The localization to the spindle and microtubule tips is very weak, suggesting that CLASP2 mediates its effects primarily at the kinetochore. The authors describe the different domains of CLASP2 as independently contributing. **In the context of the spindle, the kinetochore pool seems to be most important, so this might be clarified and discussed.**

R: We have now significantly extended our Discussion and highlighted that CLASP2 role in the spindle depends essentially (but not only) on CLASP2 at kinetochores. The new CLASP2 Δ C-Spc25 mutant, which targets CLASP2 to kinetochores as a monomer, fully rescues kinetochore microtubule half-life and flux in the absence of endogenous CLASP1/2. However, it only partially rescues spindle length, mitotic duration and segregation errors.

Cells with the various mutations show defects in chromosome segregation but the nature of the defects is not clear. Movies show chromosomes that lag, but no analysis of the types of segregation defects was provided.

R: We now provide this analysis (Please see our response to reviewer #2)

Microtubule turnover is measured in cells expressing PA-GFP tubulin. The results show that the half-time for turnover is faster in cells expressing the mutant CLASP2 proteins and depleted of endogenous CLASPs, with a more significant change when the two C-terminal mutant versions of CLASP2 are expressed. The authors say that the **cells were treated with MG132 for 30min, so they should be in metaphase**. However, prior work has shown that **kinetochore microtubule turnover is faster in prometaphase cells and becomes reduced after the last chromosome has aligned at the metaphase plate (Kebeche and Compton; please add the citation)**. **Could the change in turnover result because the cells expressing mutant CLASP take longer to progress to metaphase, and therefore are not in full metaphase when the measurements are performed? The authors should measure turnover in PM cells as well as metaphase cells and determine if the stage of mitosis impacts the measurements.**

R: All measurements were performed in comparable metaphase cells only (with fully aligned chromosomes, as verified by DIC prior to imaging). This was necessary to standardize conditions and

increase the robustness of the assay to draw meaningful comparisons between all the different mutants. Repeating all measurements for all mutant constructs in prometaphase would not be feasible during the allocated revision period (please note that we have measured microtubule half-life parameters in over 500 cells, when normally people do less than 10 per condition). This would take about a year of intense work to complete and it is uncertain whether the retrieved information will be useful given the higher variability among prometaphase cells. We also highlight that the effect of CLASP2 depletion in prometaphase was already shown to destabilize attachments (Maffini et al., CB, 2009), clearly indicating that CLASP2 has a dual role depending on the mitotic stage. We now acknowledge prior work demonstrating different kinetochore-microtubule half-life between prometaphase and metaphase. Finally, we thought that including data for metaphase cells for three new CLASP2 mutants was going to be more informative and therefore decided to concentrate our efforts in these experiments.

A related concern is that the PA-tubulin marks in **some of the cells appear to lack bright Kinetochore fiber microtubule bundles, suggesting that k-fibers do not form when CLASP is mutant/absent.** The cells should be stained for a checkpoint protein to determine if they generate stable attachments, and satisfy the checkpoint. The photoactivation data are fit to a double exponential which reports on the k-fiber and non-K microtubule populations; however **if kinetochore fibers do not form or are altered how are the data analyzed-- are both fast and slow populations present? Representative graphs of fluorescence dissipation should be shown, and the authors should provide a Table with the values for the percent of the Photoactivated microtubules that are in the fast and slow populations, and the half-time values. The rates of flux should also be provided, as it would be of interest to know if that parameter is also altered.**

R: We now include cold-shock and Mad1-staining experiments for all CLASP2 mutant conditions and compare it to Ndc80 depletion in metaphase-like cells. Our data shows that cold-stable k-fibers are clearly present in all the mutants. They are however less robust, consistent with a role for CLASP2 in the stabilization of kinetochore-microtubule attachments on bi-oriented chromosomes. Mad1-staining experiments also show that kinetochores ultimately attach to sufficient microtubules that satisfy the SAC, despite of a delay. Our FDAPA data clearly shows the presence of the two populations in all experimental conditions with a variability that was less than 15% relative to controls. This is now provided in Figure S5, together with representative graphs showing the two-populations for all experimental conditions. Finally, we have also measured poleward flux in the very same cells used to quantify microtubule half-life for all

experimental conditions. Together our results indicate that CLASP2 integrates critical microtubule-binding properties at the kinetochore-microtubule interface that sustain microtubule growth required for poleward flux and mediate kinetochore-microtubule attachment stability. This is now fully discussed in the revised manuscript.

Other:

In figure 2 the intensity of kinetochore CLASP2 appears lower in the cells expressing IP12 mutant CLASP, as compared to the CLASP with mutations in TOG domains (2ea-3eaaa) or CLASP2 with mutations in both SxIP and TOG domains. Is the image representative?

R: The reviewer is correct and indeed the intensity of kinetochore CLASP2 is lower the indicated mutants. We believe this is due to some variability in the expression of the different constructs or due to alterations in protein folding. This is now discussed in the revised manuscript.

There is no quantification of the western blots. The cell line expressing IP12-3eaaa mutant CLASP2 has (apparently) a much reduced level of protein. This is worrisome because the lower level of protein could contribute to the phenotype in these cells.

R: We now provide quantification of the western blots in Figure S2. We show that for the IP12-3eaaa mutant there is approximately 15% reduction in the protein levels and less than 20% reduction at kinetochores relative to the WT construct. Given that mitosis is essentially normal after 50% reduction in overall CLASP1 and CLASP2 proteins levels (Pereira et al., MBoC, 2006), it is unlikely that the observed variability in the expression of the different mutant constructs accounts for the reported phenotypes. This is now clarified in the Discussion section of the revised manuscript.

Supplemental figure 3. Imaging CLASP at microtubule plus-ends is weak. The cells expressing wild-type CLASP do show co-localization, but in all of the other panels, the CLASP images look very punctate without clear "dashes" (perhaps with the exception of the "dash" that is highlighted in the box insets. Even in the nuclear area, where the background is low, the colocalization is not obvious in the images supplied. This makes the evaluation of the mutants very challenging. **Can the authors either provide better images or**

discuss the evidence showing that CLASP2 tip-tracks in mitosis?

R: Full characterization of the microtubule plus-end-binding capacity of all CLASP2 microtubule-binding mutants used in the present study has been performed before in interphase cells (Maki et al., J. Mol Biol, 2015) and our findings were fully consistent with what was reported. It is also known that CLASP2 microtubule plus-end-binding is negatively regulated (but not abolished!) during mitosis and we now include these references in our extended discussion.

Similarly, **in supplemental figure 3B, the spindle staining is very weak - hardly above background, making the data hard to evaluate. The 2ea-3eeaa expressing cells have weak spindle fluorescence, but also fewer microtubules in the tubulin channel. It is very hard to say if the IP12-3eeaa mutant has any specific staining at all.**

R: We agree with the reviewer that the spindle staining, including the microtubules, is far from perfect, but this was due to the methanol fixation required to preserve the mRFP-CLASP2 signal at kinetochores. We also draw attention that mRFP is not particularly bright and we are detecting the signal directly, without any signal amplification step by secondary antibodies. The provided images are as representative as they can be.

Page 14: the authors discuss figure 5, which shows the PA of tubulin and measurement of turnover half-time. They state that the change in dynamic turnover was "exacerbated by the combined mutation of the TOG3 and SxIP domains" but in Fig 5B, the turnover of Kinetochores microtubules does not appear to be different for the double mutant, as compared to either of the constructs with a single mutation.

R: The exact numbers are now provided in Figure S5B and indeed reveal a slight reduction in kinetochore-microtubule half-life in the combined mutants of either TOG domains, particularly in the new mutant 2ea-IP12 now included in this revised manuscript.

Movie 1 siRNA CLASP2 is the chromosome lagging or did it never attach?

R: For the sake of clarity we have replaced the movie illustrating CLASP2 siRNA for a new one.

Movie 6: **is the spindle rocking/oscillating typical for cells expressing this mutant?**

R: Not really, sometimes we observe this rocking even in control cells and also in other mutants, especially when mitosis is slightly delayed.

Consider **changing incapacity to inability**.

R: The text has been changed according to the reviewer's suggestion.

Golden standard should be **gold standard**.

R: This has been corrected in the revised text.

October 29, 2019

RE: JCB Manuscript #201905080R

Prof. Helder Maiato
Institute for Molecular and Cell Biology University of Porto
Rua Alfredo Allen, 208
Porto 4200-135
Portugal

Dear Prof. Maiato,

Thank you for submitting your revised manuscript entitled "CLASP2 binding to curved microtubule tips promotes flux and stabilizes kinetochore attachments". You will see that the returning reviewers appreciated your efforts to address their points and find the revised manuscript stronger and clearer. Both referees recommend publication and suggest final edits to the text and figures that we strongly encourage you to consider and address. No further experimentation is needed. We would be happy to publish your paper in JCB pending final revisions necessary to meet our formatting guidelines (see details below) and pending revisions to address the reviewers' comments. Please include a point-by-point response to the reviewers' remaining points at resubmission.

1) eTOC summary: A 40-word summary that describes the context and significance of the findings for a general readership should be included on the title page. The statement should be written in the present tense and refer to the work in the third person.

- Please include a summary statement on the title page of the resubmission. It should start with "First author name(s) et al..." to match our preferred style.

2) Figure formatting: Scale bars must be present on all microscopy images, including inset magnifications.

- Please add scale bars to S4 magnifications.

Molecular weight or nucleic acid size markers must be included on all gel electrophoresis.

- Please add molecular weight with unit labels on the following panels: all S2 blots (including unit labels)

- Unit labels are needed for all blots (e.g., figure 1)

3) Statistical analysis: Error bars on graphic representations of numerical data must be clearly described in the figure legend. The number of independent data points (n) represented in a graph must be indicated in the legend. Statistical methods should be explained in full in the materials and methods. For figures presenting pooled data the statistical measure should be defined in the figure legends.

4) Materials and methods: Should be comprehensive and not simply reference a previous publication for details on how an experiment was performed. Please provide full descriptions in the

text for readers who may not have access to referenced manuscripts.

- Please include database IDs/vendor IDs for all constructs and cell lines, even if described in other work previously or gifted by other investigators (Addgene, ATCC, etc). If database/vendor IDs are not available, please describe their basic genetic features ****even if the vectors were described elsewhere previously**** (e.g. more info is needed about the CSII-CMV-MCS vector and its derivatives)

- Please include species, catalog IDs for all antibodies.

- Microscope image acquisition: The following information must be provided about the acquisition and processing of images:

a. Make and model of microscope

b. Type, magnification, and numerical aperture of the objective lenses

c. Temperature

d. imaging medium

e. Fluorochromes

f. Camera make and model

g. Acquisition software

h. Any software used for image processing subsequent to data acquisition. Please include details and types of operations involved (e.g., type of deconvolution, 3D reconstitutions, surface or volume rendering, gamma adjustments, etc.).

5) References: There is no limit to the number of references cited in a manuscript. References should be cited parenthetically in the text by author and year of publication.

- Please abbreviate the names of journals according to PubMed.

- Please note our reference formatting guidelines for preprints and please reformat ****both the in-text and reference list citation**** to the following paper accordingly:

Tipton, A.R., and G.J. Gorbsky. 2019. Complexities of Microtubule Population Dynamics within the Mitotic Spindle. bioRxiv:769752.

Ref guidelines: <http://jcb.rupress.org/reference-guidelines>

A. MANUSCRIPT ORGANIZATION AND FORMATTING:

Full guidelines are available on our Instructions for Authors page, <http://jcb.rupress.org/submission-guidelines#revised>. ****Submission of a paper that does not conform to JCB guidelines will delay the acceptance of your manuscript.****

B. FINAL FILES:

-- High-resolution figure and video files: See our detailed guidelines for preparing your production-ready images, <http://jcb.rupress.org/fig-vid-guidelines>.

-- Cover images: If you have any striking images related to this story, we would be happy to consider them for inclusion on the journal cover. Submitted images may also be chosen for highlighting on the journal table of contents or JCB homepage carousel. Images should be uploaded

as TIFF or EPS files and must be at least 300 dpi resolution.

****It is JCB policy that if requested, original data images must be made available to the editors. Failure to provide original images upon request will result in unavoidable delays in publication. Please ensure that you have access to all original data images prior to final submission.****

****The license to publish form must be signed before your manuscript can be sent to production. A link to the electronic license to publish form will be sent to the corresponding author only. Please take a moment to check your funder requirements before choosing the appropriate license.****

Thank you for this interesting contribution, we look forward to publishing your paper in the Journal of Cell Biology.

Sincerely,

Rebecca Heald, Ph.D.
Monitoring Editor, Journal of Cell Biology

Melina Casadio, Ph.D.
Senior Scientific Editor, Journal of Cell Biology

Reviewer #2 (Comments to the Authors (Required)):

The revised manuscript by Girão et al. has cleared up all confusion/concerns outlined in my previous comments. This work demonstrates that Clasp2's EB binding activity and TOG2 and TOG3 domains are required at kinetochores in order to stabilize metaphase microtubule attachments and exit mitosis. Interestingly, self-association and the TOG1 domain (and thus binding soluble tubulin dimers) are dispensable for this activity. With these additions to the manuscript I strongly believe it should be published in JCB. Below are a few minor comments concerning the text that the authors may want to address but by no means are a requirement for publication.

Minor Comments

- I understand that you emphasize Spc25 as a monomer, meaning it does not homo-dimerize which is important for your mutant. However, as Spc25 heterodimerizes with Spc24 it was initially a bit confusing. You may consider emphasizing the homodimerization point rather than referring to it as a monomer.
- You mention that the Clasp2-Spc25 fusion localizing to kinetochores at twice the intensity of WT is consistent with "constitutive kinetochore localization" which implies that Clasp2 kinetochore localization is more dynamic than other kinetochore factors like the CCAN. If that were the case, I would expect the WT protein to have a bimodal or at least a larger range of kinetochore levels, but its distribution is very similar to the Spc25 fusion. Instead to me it suggests something about the stoichiometry of Clasp2 at kinetochores relative to Spc25.

- On page 20 you may want to delete the repeated word 'flux'
 - o "...sufficient to ensure normal poleward flux and kinetochore-microtubule flux stability, it is critical for normal kinetochore-microtubule dynamics."

Reviewer #3 (Comments to the Authors (Required)):

Girao and co-authors have submitted a revised version of their manuscript, "CLASP2 binding to curved microtubule tips promotes flux and stabilizes kinetochore attachments". The revisions are extensive and the authors have addressed all of the concerns that I raised in the first review.

I found the revised version to be greatly improved in many ways; the new biochemical characterization, and the addition of a separate discussion and a clear introduction make the manuscript much more accessible. I recommend publication.

I have a minor issue that the authors can consider:

In Figure 4 the quantification is shown as a bar graphs, but in the other figures, the individual data points are shown. Showing the individual points for all the figures would be an improvement and would be consistent.

The images at the bottom of figure 6 could be moved to supplemental.

Point-by-point response to the reviewers (in blue)

Reviewer #2 (Comments to the Authors (Required)):

The revised manuscript by Girão et al. has cleared up all confusion/concerns outlined in my previous comments. This work demonstrates that Clasp2's EB binding activity and TOG2 and TOG3 domains are required at kinetochores in order to stabilize metaphase microtubule attachments and exit mitosis. Interestingly, self-association and the TOG1 domain (and thus binding soluble tubulin dimers) are dispensable for this activity. With these additions to the manuscript I strongly believe it should be published in JCB. Below are a few minor comments concerning the text that the authors may want to address but by no means are a requirement for publication.

R: We thank the reviewer for the continuous encouragement and for recommending our manuscript for publication in JCB. We are happy to know that we were able to clarify all previous concerns.

Minor Comments

- I understand that you emphasize Spc25 as a monomer, meaning it does not homo-dimerize which is important for your mutant. However, as Spc25 heterodimerizes with Spc24 it was initially a bit confusing. You may consider emphasizing the homodimerization point rather than referring to it as a monomer.

R: The reviewer is absolutely correct with the fact that Spc25 heterodimerizes with Spc24. However, in our fusion experiments, because only Spc25 is fused to CLASP2, it ensures that this chimeric protein will not be able to homodimerize with additional CLASP2 monomers, regardless of any potential heterodimerization with Spc24 (which lacks fused CLASP2). We have re-written the relevant section in the text to clarify this point.

- You mention that the Clasp2-Spc25 fusion localizing to kinetochores at twice the intensity of WT is consistent with "constitutive kinetochore localization" which implies that Clasp2 kinetochore localization is more dynamic than other kinetochore factors like the CCAN. If that were the case, I would expect the WT protein to have a bimodal or at least a larger range of kinetochore levels, but its distribution is very similar to the Spc25 fusion. Instead to me it suggests something about the stoichiometry of Clasp2 at kinetochores relative to Spc25.

R: This is a good point and we have now included this possibility in our explanation for the increased levels of CLASP2-Spc25 at kinetochores in the main text.

- On page 20 you may want to delete the repeated word 'flux'
o "...sufficient to ensure normal poleward flux and kinetochore-microtubule flux stability, it is critical for normal kinetochore-microtubule dynamics."

R: We have corrected the sentence.

Reviewer #3 (Comments to the Authors (Required)):

Girao and co-authors have submitted a revised version of their manuscript, "CLASP2 binding to curved microtubule tips promotes flux and stabilizes kinetochore attachments". The revisions are extensive and the authors have addressed all of the concerns that I raised in the first

review.

I found the revised version to be greatly improved in many ways; the new biochemical characterization, and the addition of a separate discussion and a clear introduction make the manuscript much more accessible. I recommend publication.

R: We thank the reviewer for recognizing our effort in improving our manuscript and for recommending its publication in JCB.

I have a minor issue that the authors can consider:

In Figure 4 the quantification is shown as a bar graphs, but in the other figures, the individual data points are shown. Showing the individual points for all the figures would be an improvement and would be consistent.

The images at the bottom of figure 6 could be moved to supplemental.

R: We thank the reviewer for the suggestions, but in figure 4 we are representing in a bar graph the frequency of a particular event as a percentage of cells, rather than the distribution of the individual data points when measuring a specific aspect as in other figures. We also could not move the images at the bottom of figure 6 to supplemental because we are already at the limit of supplementary figures allowed by JCB.